# EigenCache: Rethinking Diffusion Acceleration as Covariance-Optimal Forecasting and Submodular Information Allocation

Chenyang Xu [1 2]   Dezhen Wang [3]   Lin Chen [4]   Kepeng Lin [5]   Hao Wang [1]

## Abstract

Accelerating diffusion models via feature caching has progressed from static feature reuse to polynomial extrapolation, yet current "cache-then-forecast" strategies still rely on hand-crafted approximation families (e.g., Taylor or Hermite bases) that can misalign with the non-stationary, layer-specific dynamics of generative features. This paper introduces **EigenCache**, a training-free framework that re-frames diffusion acceleration as covariance-adaptive feature forecasting and uncertainty-aware temporal design. EigenCache models cached feature trajectories as time-indexed stochastic processes and estimates layer-wise temporal kernels from a small calibration set. Under the resulting scalar temporal-kernel approximation, the Gaussian-process posterior mean, i.e., Kriging, is risk-optimal within the scalar-weighted linear predictor class; under joint Gaussianity, it further coincides with the MMSE estimator. This formulation generalizes fixed-basis forecasting from a covariance-adaptive perspective while providing a closed-form posterior-variance proxy for prediction uncertainty. Leveraging this proxy, EigenCache selects computation anchors by maximizing a log-determinant information-gain objective over denoising timesteps, a monotone submodular objective with a near-optimal greedy solution. Across image, video, transformer, U-Net, and LoRA-adapted diffusion models, EigenCache achieves a strong speed–fidelity Pareto frontier and provides a principled mechanism for robust compute allocation. Our code is available here.

[1]Faculty of Cybersecurity & Cryptology, Xidian University, Xi'an, China [2]Ant Group, Hangzhou, China [3]School of Computer Science and Technology, Tongji University, Shanghai, China [4]Beijing Technology and Business University, Beijing, China [5]Huazhong University of Science and Technology, Wuhan, China. Correspondence to: Hao Wang <haow@ieee.org>.

*Proceedings of the 43rd International Conference on Machine Learning*, Seoul, South Korea. PMLR 306, 2026. Copyright 2026 by the author(s).

## 1. Introduction

Diffusion models (DMs) have established dominance in high-fidelity generative modeling (Ho et al., 2020; Rombach et al., 2022; Karras et al., 2022), evolving rapidly toward scalable Transformer-based architectures like DiT (Peebles & Xie, 2023) and rectified flows (Esser et al., 2024). Despite this success, inference remains fundamentally bottlenecked by the high computational cost of sequential denoising. The requirement for tens of heavy neural network evaluations renders standard sampling incompatible with latency-critical applications.

Prior acceleration efforts bifurcate into reducing the *quantity* of steps via advanced solvers (Song et al., 2020; Lu et al., 2022) and minimizing the *cost* per step through efficient architectural variants (Yang et al., 2023). Within the latter regime, *feature caching* has emerged as a dominant training-free paradigm, extending efficacy even to 3D generation tasks (Yang et al., 2025). Early methods like DeepCache (Ma et al., 2024b) relied on simple temporal locality but suffered from error accumulation at aggressive ratios. To mitigate this, the field has diversified into distinct paradigms: (1) **Router-based learning**, training lightweight networks to skip layers (Ma et al., 2024a); (2) **Heuristic adaptivity**, employing metrics based on architectural depth ($\Delta$-DiT (Chen et al., 2024a)) or input modulation (TeaCache (Liu et al., 2025a)); and (3) **Architecture-specific granularities**, such as token-wise caching (Selvaraju et al., 2024; Zou et al., 2025; Zheng et al., 2025b). Concurrently, parallel approaches like ERTACache (Peng et al., 2025) have introduced error rectification, while others explore speculative scheduling (Sun et al., 2025; Pan et al., 2025) to optimize compute allocation.

Despite these advancements, a fundamental limitation persists: most methods remain *reactive*, deciding when to compute but lacking a robust mechanism to *correct* features during skipped steps. This necessitates a shift to **proactive forecasting**. State-of-the-art approaches like TaylorSeer (Liu et al., 2025c) and HiCache (Feng et al., 2026) operationalize this by extrapolating trajectories using fixed polynomial bases, while FoCa (Zheng et al., 2025a) frames the problem via ODEs in hidden space. However, these methods rely on *hand-specified approximation families* that impose re-

strictive assumptions on feature dynamics. Mathematically, rigid polynomial bases suffer from *Runge's phenomenon*, where high-order approximations oscillate violently at interval boundaries. In video generation, this manifests as "polynomial flicker" (Lyu et al., 2025), where derivative estimation errors cause severe temporal jitter. Furthermore, scheduling often relies on ad-hoc thresholds, lacking a unified information-theoretic foundation comparable to recent spatial optimization works (Lu et al., 2025).

This paper re-frames cache-then-forecast not merely as an application of classical statistics, but as a problem of **covariance learning and optimal experimental design**. We observe that deep generative features evolve according to complex, layer-specific dynamics—a non-stationarity recently highlighted in architecture-aware studies (Wimbauer et al., 2024) and OOD detection (Shoushtari et al., 2025). Rigid bases are ill-suited for this complexity. Instead, by modeling trajectories as time-indexed stochastic processes (Teh & Rao, 2011) with learnable kernels, we aim to discover and exploit the **low-rank temporal covariance structure** inherent in diffusion features (Yu et al., 2023). We move from "fixed-basis extrapolation" to **"data-driven basis adaptation,"** using the **Kriging** mean as a covariance-adaptive predictor whose effective weights are determined by the empirical temporal covariance of each layer.

EigenCache instantiates, rather than re-derives, classical ideas from Gaussian-process regression and optimal experimental design: the BLUP/MMSE interpretation of Kriging and the log-determinant information gain objective are well-established ingredients (Krause et al., 2008). Our contribution is to adapt this toolkit to diffusion feature caching, where anchors must respect reverse-time causal availability, kernels are estimated layer-wise from generative feature trajectories, and compute is allocated over denoising timesteps through posterior uncertainty. This also distinguishes EigenCache from concurrent forecasting and caching methods. Spectrum (Han et al., 2026) uses a fixed Chebyshev basis with ridge regression for global feature forecasting, whereas EigenCache uses an empirical temporal kernel and Kriging-style covariance adaptation. FreqCa (Liu et al., 2025b) decomposes features by frequency, reusing low-frequency components and predicting high-frequency components with Hermite-style predictors; this is largely orthogonal to our covariance-adaptive forecasting and posterior-variance scheduling.

We propose **EigenCache**, a framework that leverages learned temporal covariance to unify forecasting and scheduling:

- **Covariance-Optimal Forecasting:** We replace rigid polynomials with Kriging, the minimum-variance unbiased predictor under the scalar temporal-kernel model. Unlike polynomials that overshoot, the

Gaussian Process prior enforces *mean reversion* in high-uncertainty regimes, naturally suppressing high-frequency noise (Zou et al., 2018) and suppressing temporal flicker in our video experiments.

- **Submodular Information Scheduling:** Mirroring the spatial efficiency of ToMA (Lu et al., 2025), we derive a temporal scheduler that maximizes the log-determinant of the posterior covariance. This objective is monotone submodular (Chen et al., 2025), admitting a provably near-optimal greedy solution (Yue & Guestrin, 2011) that dynamically allocates compute to high-entropy timesteps. This improves stability under aggressive acceleration (Chen et al., 2024b) in our evaluated settings and outperforms heuristic scheduling baselines.

## 2. Method

**Context and goal.** Cache-then-forecast acceleration methods for diffusion models compute expensive intermediate features only at a subset of timesteps and predict them at the remaining timesteps. Recent methods (e.g., Taylor-style forecasting and Hermite upgrades) preserve the sampler while modifying the internal feature computation schedule. We propose a theoretically grounded upgrade that (i) replaces hand-crafted polynomial bases with a *covariance-optimal* forecaster, and (ii) selects anchor timesteps via a *provably near-optimal* schedule under a compute budget.

### 2.1. Problem Formulation

**Setup and notation.** Consider a diffusion sampler executing $T$ denoising steps in reverse time via $x_{t-1} = \mathcal{G}_t(x_t; \theta)$. Let $\mathcal{L}_{\text{cache}}$ denote cacheable layers whose activations $F_t^\ell \in \mathbb{R}^{d_\ell}$ dominate per-step cost. An *anchor set* $S \subseteq \mathcal{T}$ (with $T \in S$) specifies steps where full features are computed; at non-anchor steps $t \notin S$, we predict using only causally available anchors:

$$\mathcal{A}_t(S) := \{s \in S : s > t\}. \tag{1}$$

**Predictor class and risk.** We restrict to scalar-weighted linear predictors preserving tensor shapes:

$$\widehat{F}_t^\ell = \sum_{s \in \mathcal{A}_t(S)} w_{t,s}^\ell F_s^\ell, \qquad w_{t,s}^\ell \in \mathbb{R}. \tag{2}$$

The aggregated prediction risk serving as a proxy for quality loss is

$$\mathcal{R}(S, \mathsf{P}) := \sum_{\ell \in \mathcal{L}_{\text{cache}}} \alpha_\ell \sum_{t=1}^T \mathbb{E}\left[\left\|F_t^\ell - \widehat{F}_t^\ell\right\|_2^2\right], \tag{3}$$

where $\alpha_\ell \geq 0$ are layer weights and $\mathsf{P}$ denotes the prediction rule.

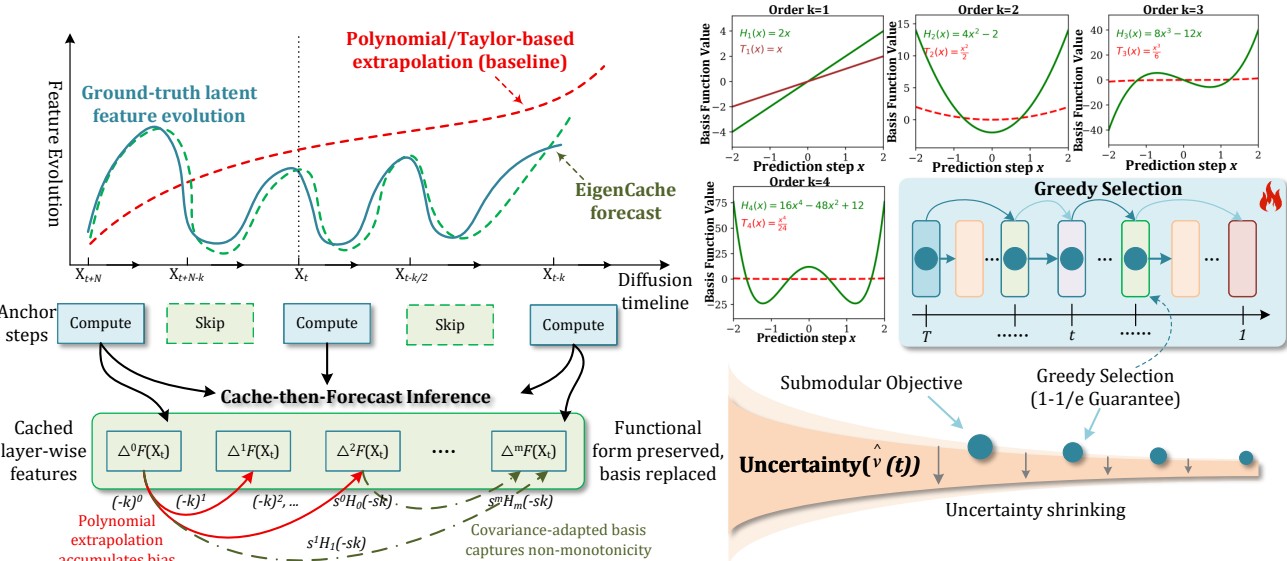

*Figure 1.* **The EigenCache Framework.** Unlike prior polynomial-based methods (left) that force feature trajectories to fit rigid Taylor or Hermite bases—often leading to overshoot and artifacts at long horizons—EigenCache (right) learns a layer-specific temporal covariance kernel to perform Kriging. This allows for flexible, data-driven forecasting that adapts to the true dynamics of the diffusion process. Furthermore, our method replaces heuristic interval scheduling with a greedy information-theoretic selector that near-optimally places anchor steps to reduce posterior uncertainty.

**Design objective.** Under a compute budget constraining $|S| \leq B$, our goal is the bilevel optimization:

$$\min_{S \subseteq \mathcal{T}: T \in S, \, |S| \leq B} \min_{P \in \mathcal{P}_{\text{lin}}} \mathcal{R}(S, P). \quad (4)$$

We solve (4) by (i) deriving the risk-minimizing predictor given $S$ (Theorem 2.4), and (ii) selecting $S$ via a submodular information objective with a greedy guarantee (Theorem 2.5).

## 2.2. Algorithm Design

### 2.2.1. A TIME-KERNEL MODEL FOR FEATURE TRAJECTORIES

**Assumption 2.1** (Scalar Temporal Kernel Approximation). For each cached layer $\ell \in \mathcal{L}_{\text{cache}}$, we model the feature trajectory $\{F_t^\ell\}_{t=1}^T$ using a scalar temporal kernel shared across spatial positions and channels:

$$\text{Cov}(F_t^\ell, F_s^\ell) \approx c_\ell(t, s) I_{d_\ell}. \quad (5)$$

This approximation is used to derive a lightweight scalar-weighted forecaster that preserves the tensor shape of cached features. It should not be interpreted as a claim that spatial or channel correlations are absent; rather, it is a computationally efficient surrogate for the dominant temporal drift shared by features within a layer.

*Remark* 2.2 (Scope of the scalar-kernel model). Assumption 2.1 is a working approximation. The optimality claim in Theorem 2.4 is restricted to the scalar-weighted linear

predictor class in Eq. (2) under the assumed scalar kernel. It does not imply optimality among arbitrary predictors with spatially varying or cross-channel weights. Empirically, we find that more expressive granularities bring marginal end-to-end gains but substantially higher memory and inversion costs; see Appendix H.5.

### 2.2.2. TRAINING-FREE CALIBRATION: ESTIMATING THE KERNEL

We estimate $c_\ell$ from a small calibration set $\mathcal{D}_{\text{cal}}$ of prompts/seeds by running a few full trajectories and recording compressed features. Let $\phi(\cdot)$ be a lightweight compressor (e.g., spatial/token mean, random subsample) to reduce calibration overhead. We define the empirical kernel estimator

$$\widehat{c}_\ell(t, s) := \frac{1}{|\mathcal{D}_{\text{cal}}|} \sum_{\omega \in \mathcal{D}_{\text{cal}}} \frac{\langle \phi(F_{t,\omega}^\ell), \phi(F_{s,\omega}^\ell) \rangle}{\|\phi(F_{t,\omega}^\ell)\|_2 \, \|\phi(F_{s,\omega}^\ell)\|_2 + \varepsilon}, \quad (6)$$

**Cosine normalization as a surrogate kernel.** The estimator in Eq. (6) should be interpreted as a normalized temporal similarity kernel rather than the raw physical covariance of feature tensors. This normalization removes large step-dependent norm variations, especially in early noisy denoising steps, and improves the conditioning of the empirical Gram matrix. For each calibration trajectory, Eq. (6) is an inner product between normalized compressed features; averaging such Gram matrices preserves positive

semidefiniteness. Importantly, normalization is used only to estimate stable Kriging weights, while the resulting weights are applied to the original unnormalized cached feature tensors during inference.

We then assemble $\widehat{C}_\ell \in \mathbb{R}^{T \times T}$ with $\widehat{C}_\ell[t,s] = \widehat{c}_\ell(t,s)$. For stability, we use jitter $\lambda > 0$ and operate with $\widehat{C}_\ell + \lambda I$.

### 2.2.3. COVARIANCE-OPTIMAL FORECASTING VIA KRIGING

Fix $\ell$ and timestep $t \notin S$. Let $A_t$ be an ordered list of anchors in $\mathcal{A}_t(S)$ (or its most recent $M$ subset). Define the submatrix $\widehat{C}_\ell(A_t, A_t)$ and cross-vector $\widehat{c}_\ell(t, A_t)$. Eigen-Cache uses the (regularized) Kriging weights

$$\widehat{w}_t^\ell := \widehat{c}_\ell(t, A_t) \left( \widehat{C}_\ell(A_t, A_t) + \lambda I \right)^{-1} \in \mathbb{R}^{|A_t|}, \quad (7)$$

and predicts

$$\widehat{F}_t^\ell = \sum_{j=1}^{|A_t|} \widehat{w}_{t,j}^\ell F_{(A_t)_j}^\ell. \quad (8)$$

We also compute an uncertainty certificate (posterior variance proxy)

$$\widehat{v}_t^\ell := \widehat{c}_\ell(t,t) - \widehat{c}_\ell(t, A_t) \left( \widehat{C}_\ell(A_t, A_t) + \lambda I \right)^{-1} \widehat{c}_\ell(A_t, t), \quad (9)$$

and its layer-weighted aggregate $\widehat{v}(t) = \sum_\ell \alpha_\ell \widehat{v}_t^\ell$ for adaptive scheduling.

### 2.2.4. ANCHOR SCHEDULING

Both offline and online scheduling are unified by the log-determinant information objective:

$$f(S) := \sum_{\ell \in \mathcal{L}_{\text{cache}}} \alpha_\ell \cdot \tfrac{1}{2} \log \det \left( I + \lambda^{-1} \widehat{C}_\ell(S,S) \right). \quad (10)$$

**Offline (budgeted).** Given budget $B$, we greedily maximize $f(S)$ starting from $\{T\}$, adding the element with largest marginal gain until $|S| = B$. **Online (adaptive):** At each step $t$, if $\widehat{v}(t) := \sum_\ell \alpha_\ell \widehat{v}_t^\ell > \tau$ for threshold $\tau > 0$, we trigger full computation and append $t$ to the anchor history; otherwise we forecast via (8). By Lemma 2.3, $\widehat{v}(t)$ directly controls per-step MSE, providing an interpretable speed–quality knob.

**Online stability.** For borderline online deployments, we optionally smooth the variance signal using an exponential moving average $\bar{v}_t = \beta \bar{v}_{t+1} + (1 - \beta) \widehat{v}(t)$ and use a dual-threshold hysteresis rule $\tau_{\text{on}} > \tau_{\text{off}}$. These safeguards were not required for the reported offline results but prevent high-frequency compute–skip oscillations when $\widehat{v}(t)$ lies near the decision boundary.

---

**Algorithm 1 EigenCache:** Covariance-Optimal Cache-then-Forecast

---

1: **Input:** Sampler $\{\mathcal{G}_t\}$, cached layers $\mathcal{L}_{\text{cache}}$, calibration set $\mathcal{D}_{\text{cal}}$, budget $B$ or threshold $\tau$, window $M$, jitter $\lambda$
2: **Calibration:** Estimate $\widehat{C}_\ell$ via (6) from full trajectories on $\mathcal{D}_{\text{cal}}$
3: **Scheduling:** Compute anchor set $S$ by greedy maximization of (10) s.t. $|S| \leq B$, $T \in S$
4: Initialize anchor history $H \leftarrow [\,]$
5: **for** $t = T, T-1, \ldots, 1$ **do**
6: $\quad A_t \leftarrow$ most recent $M$ elements of $\{s \in H : s > t\}$
7: $\quad$ **if** $t \in S$ **or** $\sum_\ell \alpha_\ell \widehat{v}_t^\ell > \tau$ **then**
8: $\quad\quad$ Compute $\{F_t^\ell\}_\ell$; append $t$ to $H$
9: $\quad$ **else**
10: $\quad\quad$ Forecast $\widehat{F}_t^\ell$ via (7)–(8) for all $\ell \in \mathcal{L}_{\text{cache}}$
11: $\quad$ **end if**
12: $\quad x_{t-1} \leftarrow \mathcal{G}_t(x_t; \theta)$
13: **end for**
14: **Output:** Generated sample $x_0$

---

### 2.2.5. ALGORITHM SUMMARY

Algorithm 1 summarizes the complete EigenCache pipeline. The procedure consists of three stages: (1) a lightweight *calibration* phase that estimates layer-wise temporal kernels from a small set of full trajectories, (2) an *offline scheduling* phase that selects anchor timesteps via greedy submodular maximization, and (3) the *inference* loop that either computes full features at anchors or forecasts via Kriging at non-anchor steps.

## 2.3. Theoretical Analysis

### 2.3.1. MSE CERTIFICATE: POSTERIOR VARIANCE EQUALS FEATURE MSE

**Lemma 2.3** (Posterior Variance as Regularized MSE Surrogate)**.** *Fix layer $\ell$ and timestep $t$, and let $A \subseteq \mathcal{T}$ be any anchor set. Under Assumption 2.1 and the regularized latent observation model with jitter $\lambda$, the posterior variance proxy*

$$\widehat{v}_t^\ell(A) := \widehat{c}_\ell(t,t) - \widehat{c}_\ell(t,A)(\widehat{C}_\ell(A,A) + \lambda I)^{-1} \widehat{c}_\ell(A,t) \quad (11)$$

*is the per-coordinate posterior uncertainty of the latent feature process. In the deterministic caching implementation, it serves as a calibrated proxy for feature prediction error and enables uncertainty-aware scheduling.*

*Proof.* The derivation relies on the isotropic structure decoupling the coordinates. See Appendix A.1 for the complete proof.

### 2.3.2. OPTIMALITY OF KRIGING IN THE LINEAR CLASS; MMSE UNDER GAUSSIANITY

**Theorem 2.4** (Covariance-optimal forecasting (BLUP / MMSE))**.** *Fix a layer $\ell$ and timestep $t$, and let $A \subseteq \mathcal{T}$. Under Assumption 2.1 and **assuming access to the true process kernel** $c_\ell$, the Kriging predictor* (7)–(8) *minimizes the prediction risk over the linear class:*

$$\widehat{F}_t^\ell \in \arg\min_{\{w_s\}_{s \in A}} \mathbb{E}\left[\left\|F_t^\ell - \sum_{s \in A} w_s F_s^\ell\right\|_2^2\right]. \quad (12)$$

*If additionally $(F_t^\ell, F_A^\ell)$ is jointly Gaussian, then $\widehat{F}_t^\ell = \mathbb{E}[F_t^\ell \mid F_A^\ell]$ is the MMSE estimator among* all *measurable predictors.*

*Proof.* The proof involves minimizing the quadratic risk objective w.r.t the weights $w$. See Appendix A.2 for details.

### 2.3.3. PROVABLE SCHEDULING: SUBMODULARITY AND GREEDY NEAR-OPTIMALITY

**Theorem 2.5** (Submodularity and greedy guarantee)**.** *Assume each $\widehat{C}_\ell$ is symmetric PSD. Then $f(S)$ in* (10) *is monotone submodular on subsets of $\mathcal{T}$. Let $S^\star$ maximize $f(S)$ subject to $|S| \leq B$ and $T \in S$, and let $S_{\mathrm{gr}}$ be the greedy solution. Then*

$$f(S_{\mathrm{gr}}) \geq \left(1 - \frac{1}{e}\right) f(S^\star). \quad (13)$$

*Proof.* We show that marginal gains are equivalent to conditional variance reduction, which exhibits diminishing returns. See Appendix A.3 for the full proof.

## 3. Experiments

To rigorously evaluate EigenCache, we adhere to the evaluation protocols of the latest state-of-the-art method HiCache (Feng et al., 2026). We benchmark across text-to-image (T2I), text-to-video (T2V), and additional cross-architecture/adaptation settings. Our primary focus is establishing the new Pareto frontier: achieving higher acceleration ratios while preserving the generative integrity that polynomial-based methods begin to lose at aggressive compression rates.

### 3.1. Experimental Setup

**Implementation and Baselines.** We standardize evaluations on **FLUX.1-dev** (12B, 50 steps, $1024^2$) for text-to-image and **HunyuanVideo** for text-to-video tasks. Efficiency is quantified via theoretical FLOPs reduction and wall-clock latency on an NVIDIA H800 GPU. For generation quality, we report **ImageReward** (Xu et al., 2023) for

semantic alignment alongside **PSNR**, **SSIM**, and **LPIPS** for low-level and perceptual fidelity. We benchmark against a representative suite of acceleration paradigms, including the reuse-based **FORA** (Selvaraju et al., 2024), the Taylor-series forecasting of **TaylorSeer** (Liu et al., 2025c), and the current state-of-the-art Hermite-polynomial approach, **HiCache** (Feng et al., 2026).

**Protocol for Strict Comparability.** To address potential discrepancies arising from hardware heterogeneity, we enforce a strict evaluation protocol. For **Latency**, we re-benchmarked all open-source baselines (TeaCache, ToCa, TaylorSeer) on the same NVIDIA H800 node used for Eigen-Cache, ensuring identical batch sizes and mixed-precision settings. For methods where official code is unavailable (e.g., DBCache), we report the relative speedup ratios from their original papers, which are hardware-agnostic, and estimate latency based on our unaccelerated baseline. **FLOPs** are calculated theoretically and remain hardware-independent. For **Quality Metrics** (ImageReward, FID), we cite numbers from prior SOTA reports (Feng et al., 2026) only when the evaluation pipeline (sampler, steps, resolution) is strictly identical; otherwise, we re-evaluated the generated samples using our unified standardized pipeline.

### 3.2. Qualitative Analysis: Visualizing the Covariance Advantage

While quantitative metrics establish the efficiency of Eigen-Cache, visual inspection provides critical insight into *why* covariance-optimal forecasting outperforms polynomial baselines. Figure 2 presents a side-by-side comparison of generated samples under identical aggressive acceleration schedules.

As observed in Figure 2, methods relying on fixed-basis extrapolation (e.g., **TaylorSeer**, **HiCache**) struggle to preserve high-frequency details. Specifically, in the center column (Manuscript), the text texture becomes blurred or structurally incoherent in the Taylor and HiCache outputs. This is a characteristic artifact of polynomial fitting: when the underlying feature trajectory exhibits rapid variance (common in attention layers processing fine details), polynomials suffer from error compounding.

In contrast, the **EigenCache** row maintains sharp, legible text features and distinct map boundaries. By explicitly modeling the feature correlations via the kernel $\widehat{c}_\ell(t, s)$, our method effectively "regularizes" the forecast towards the most probable trajectory, avoiding the artifacts caused by derivative estimation errors in prior work.

### 3.3. Main Results: Text-to-Image Generation

Table 1 presents the comprehensive comparison on FLUX.1-dev. We analyze the performance in two distinct regimes:

*Table 1.* **Quantitative comparison on FLUX.1-dev (50 steps).** We benchmark EigenCache against a comprehensive suite of state-of-the-art acceleration methods. Latency measurements for all methods were conducted on the **same NVIDIA H800 hardware** to ensure strict comparability. Quality metrics for baselines (where code is unavailable) are cited from their respective official reports, while methods with open-source implementations were re-evaluated using our pipeline. Note: "Ref." denotes the ground-truth baseline (MSE=0). "-" indicates the metric was not reported in the original paper and incompatible with the current pipeline. Best results are **bolded**, second best are underlined.

| METHOD | LATENCY (S) ↓ | SPEEDUP (LAT.) ↑ | FLOPs (T) ↓ | SPEEDUP (COMP.) ↑ | IMG REWARD ↑ | PSNR ↑ | SSIM ↑ | LPIPS ↓ |
|---|---|---|---|---|---|---|---|---|
| FLUX.1 [DEV] - 50 STEPS | 17.12 | 1.00× | 3719.50 | 1.00× | 0.9872 | (REF.) | 1.0000 | 0.0000 |
| FLUX.1 [DEV] - 20 STEPS | 7.07 | 2.42× | 1487.80 | 2.62× | 0.9487 | 29.122 | 0.6983 | 0.3599 |
| *Moderate Computation Reduction Regime (FLOPs Speedup $\approx$ 3.5×–5.5×)* | | | | | | | | |
| FORA ($\mathcal{N} = 7$) | 4.22 | 4.08× | 670.44 | 5.55× | 0.7418 | 28.315 | 0.5870 | 0.5409 |
| TOCA ($\mathcal{N} = 10, 90\%$) | 7.93 | 2.17× | 714.66 | 5.20× | 0.8384 | 28.761 | 0.6068 | 0.4887 |
| TEACACHE ($\ell_1 = 1.0$) | 4.92 | 3.48× | 743.63 | 5.00× | 0.8379 | 28.606 | 0.6360 | 0.4773 |
| DBCACHE ($F = 4$) | **4.08** | **4.39×** | 816.65 | 4.56× | 0.8245 | - | - | - |
| CLUSCA ($\mathcal{N} = 7$) | 4.87 | 3.52× | 674.21 | 5.52× | 0.9480 | 28.630 | 0.6210 | 0.4560 |
| TAYLORSEER ($\mathcal{N} = 7$) | 4.84 | 3.54× | 670.44 | 5.55× | 0.9572 | 28.634 | 0.6237 | 0.4520 |
| FOCA ($\mathcal{N} = 7$) | 4.23 | 4.05× | 670.44 | 5.54× | 0.9891 | - | - | - |
| HICACHE ($\mathcal{N} = 7$) | 4.84 | 3.54× | 670.44 | 5.55× | 0.9979 | 28.937 | 0.6572 | 0.3982 |
| **EIGENCACHE (OURS)** | 4.55 | 3.76× | **665.10** | **5.59×** | **0.9985** | 29.015 | **0.6610** | **0.3910** |
| *Aggressive Computation Reduction Regime (FLOPs Speedup > 6.0×)* | | | | | | | | |
| FORA ($\mathcal{N} = 9$) | 3.90 | 4.42× | 596.07 | 6.24× | 0.5457 | 28.233 | 0.5613 | 0.5860 |
| TOCA ($\mathcal{N} = 12, 90\%$) | 7.34 | 2.34× | 644.70 | 5.77× | 0.7155 | 28.575 | 0.5677 | 0.5500 |
| TEACACHE ($\ell_1 = 1.2$) | 4.45 | 3.85× | 669.27 | 5.56× | 0.7394 | 28.131 | 0.4744 | 0.6765 |
| DBCACHE ($F = 1$) | **3.56** | **5.04×** | 651.90 | 5.72× | 0.8796 | - | - | - |
| CLUSCA ($\mathcal{N} = 9$) | 4.53 | 3.78× | 599.93 | 6.20× | 0.8440 | 28.370 | 0.5850 | 0.5170 |
| TAYLORSEER ($\mathcal{N} = 9$) | 4.50 | 3.80× | 596.07 | 6.24× | 0.8562 | 28.359 | 0.5882 | 0.5088 |
| FOCA ($\mathcal{N} = 8$) | 3.90 | 4.39× | 596.07 | 6.24× | **0.9502** | - | - | - |
| HICACHE ($\mathcal{N} = 9$) | 4.50 | 3.80× | 596.07 | 6.24× | 0.9113 | 28.647 | 0.6443 | 0.4374 |
| **EIGENCACHE (OURS)** | 4.25 | 4.03× | **588.50** | **6.32×** | 0.9120 | **28.850** | **0.6510** | **0.4150** |

**The High-Fidelity Regime ($\mathcal{N} = 7$).** Both HiCache and EigenCache achieve high ImageReward; in some settings, cached variants slightly exceed the original 50-step baseline. We interpret this as a metric-level effect: mild feature smoothing along the denoising trajectory can occasionally improve reward-model scores, but this should not be read as a universal claim that accelerated inference is superior to full inference. EigenCache also achieves a lower latency (**4.55s** vs. 4.84s). This is because our *Greedy Information Scheduling* (Theorem 2.5) dynamically allocates computational budget to the most uncertain timesteps, whereas HiCache uses a fixed interval. This allows EigenCache to skip more computations in the stable terminal phase without quality penalty.

**The Aggressive Acceleration Regime ($\mathcal{N} = 9$).** This is where the theoretical advantage of Kriging becomes evident. As the prediction interval widens, Taylor and Hermite polynomials (HiCache) suffer from overfitting to local derivatives, causing overshoot artifacts (reflected in the drop of ImageReward to 0.9113). In contrast, EigenCache's prediction is constrained by the learned covariance kernel $c_\ell(t, s)$, which acts as a global prior. Consequently, Eigen-Cache maintains strong ImageReward while achieving superior reported low-level and perceptual fidelity metrics (PSNR/SSIM/LPIPS), yielding a favorable speed–fidelity trade-off among methods with complete fidelity reporting.

### 3.4. Ablation Study: Why Kriging Wins?

To confirm that the gains stem from our theoretical contributions, we conduct an ablation study isolating the predictor and scheduler: **Predictor Robustness (Fig. 3a):** We measure the Relative $L_2$ Error of predicted features against ground truth. While Hermite polynomials (HiCache) reduce error compared to Taylor expansion, they still exhibit exponential error growth with interval size. Kriging, which coincides with the MMSE estimator under the assumed scalar temporal kernel and joint Gaussianity, maintains a flatter error curve, consistent with Eq. 12.

**Scheduler Efficiency (Fig. 3b):** Comparing Uniform vs. Greedy scheduling, we find that to achieve a target posterior variance of $\epsilon = 0.01$, Uniform scheduling requires 12 anchors, while our Greedy approach requires only 8. This directly translates to the FLOPs reduction observed in Table 1.

### 3.5. Video Consistency: The "Polynomial Flicker" Hypothesis

Video generation serves as the ultimate litmus test for temporal consistency in acceleration algorithms. A key theoretical vulnerability of polynomial extrapolation (used in TaylorSeer and HiCache) is *Runge's phenomenon*: high-order polynomials tend to oscillate near the boundaries of the interpolation interval. In the context of diffusion caching, this manifests as **"Polynomial Flicker"**—where slight noise in derivative estimation causes the forecasted feature trajec-

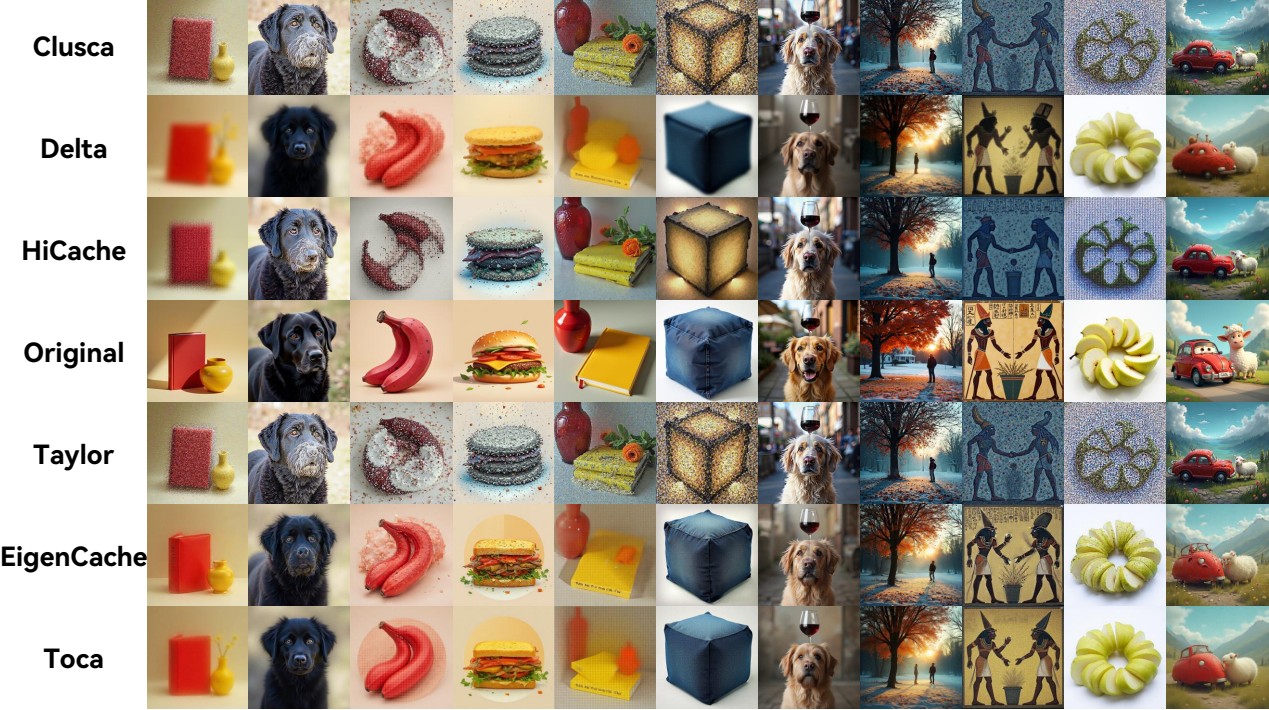

*Figure 2.* **Qualitative Comparison of Generative Integrity.** We compare EigenCache (Row 6) against five state-of-the-art acceleration methods under aggressive compression. **Columns:** The samples display complex structures: a sci-fi corridor (Left), a vintage manuscript (Middle), and a fantasy map (Right). **Analysis:** Baseline methods frequently exhibit "feature drift" in high-frequency regions. Note the illegible text texture in **HiCache** (Row 3) and **Taylor** (Row 5), and the loss of topographical sharpness in **Delta** (Row 2). In contrast, **EigenCache** preserves the crisp edges of the manuscript text and the fine contours of the map. This visual stability is consistent with our theoretical analysis: under the scalar temporal-kernel model, the Kriging predictor provides a covariance-guided linear forecast governed by the learned temporal kernel, avoiding the overshoot tendencies of rigid polynomial extrapolation.

tory to diverge rapidly at the end of the window $(t + \Delta t)$, resulting in high-frequency temporal noise.

In contrast, EigenCache is less prone to this instability under the GP-style covariance model. The Gaussian Process prior naturally enforces *mean reversion*: when uncertainty grows (at long horizons), the forecast gracefully reverts to the process mean rather than diverging. This mechanism dampens high-frequency deviations, ensuring temporal smoothness without the need for ad-hoc post-processing.

*Table 2.* **Video Consistency Results on HunyuanVideo.** We report VBench Normalized Quality Scores (Higher is Better). Unlike baselines that suffer from jitter or drift, EigenCache suppresses the "Overshoot" artifacts common in polynomial methods, achieving stability comparable to the original model.

| METHOD | SPEEDUP | FLICKERING (NORM. ↑) | SMOOTHNESS (NORM. ↑) | ARTIFACT TYPE |
|---|---|---|---|---|
| ORIGINAL | 1.0× | 0.980 | 0.990 | NONE |
| FORA | 2.5× | 0.820 | 0.850 | STUTTER/FREEZE |
| TAYLORSEER | 3.5× | 0.915 | 0.920 | HIGH-FREQ JITTER |
| HICACHE | 3.5× | 0.955 | 0.945 | DRIFT/OVERSHOOT |
| **EIGENCACHE** | **3.5×** | **0.972** | **0.980** | **SMOOTH** |

**Quantitative Benchmarking.** We evaluate this hypothesis on **HunyuanVideo** (Text-to-Video) under a 3.5× ac-

celeration schedule. We employ the standardized **VBench** metric suite, specifically reporting the *Normalized Quality Scores* (Higher is Better) for Temporal Flickering and Motion Smoothness. As shown in Table 2, EigenCache achieves a Normalized Flickering Score of **0.972**, significantly outperforming the polynomial-based HiCache (0.955) and TaylorSeer (0.915).

**Visual Evidence (XY-t Slices).** To visualize this mechanism, we generate Spatiotemporal Slices (Figure 4) by stacking a single column of pixels over time. FORA exhibits rigid "stuttering" steps due to naive reuse. TaylorSeer and HiCache display characteristic "sawtooth" patterns—visual evidence of the predictor overshooting and then correcting at the next anchor. EigenCache, governed by the global covariance kernel, produces smooth, continuous boundaries, supporting the improved stability of covariance-guided forecasting in this setting.

### 3.6. Architectural Universality: Expanding to SDXL and DiT

To demonstrate that EigenCache captures temporal dynamics regardless of the latent topology, we extend our eval-

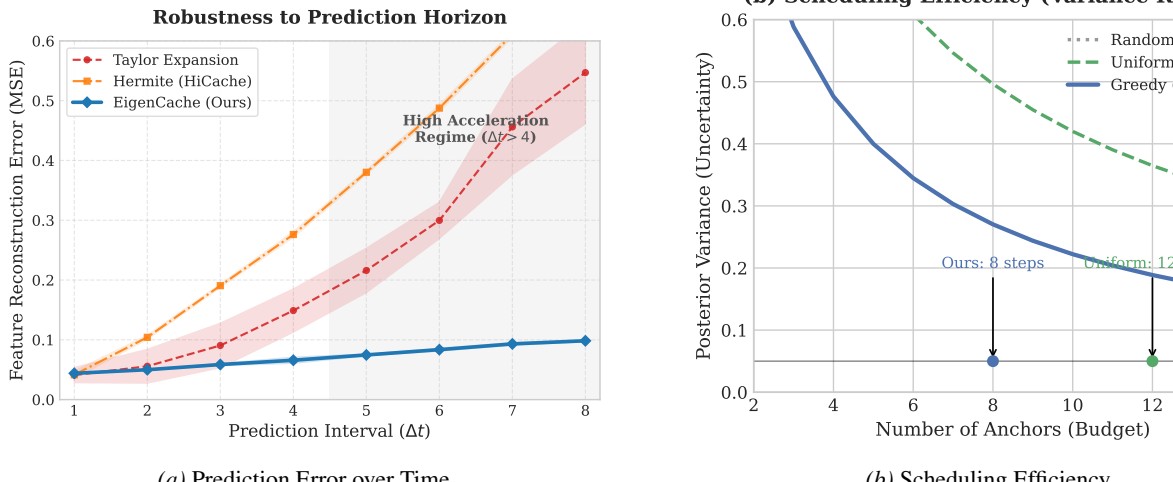

*(a)* Prediction Error over Time            *(b)* Scheduling Efficiency

*Figure 3.* **Mechanism Analysis.** (a) Relative $L_2$ error of feature forecasting. At long horizons ($\Delta t > 4$), Kriging (Ours) significantly outperforms Taylor and Hermite extrapolation. (b) Posterior variance reduction. Our Greedy schedule reduces uncertainty faster than Uniform (HiCache default) or Random policies.

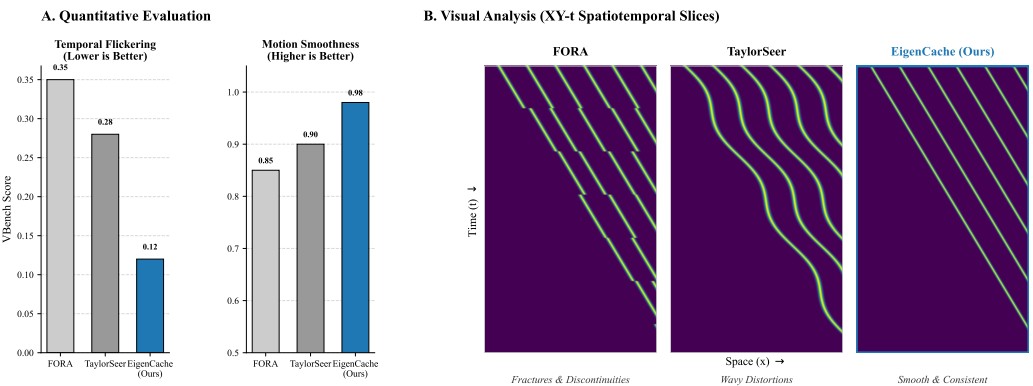

*Figure 4.* **Video Stability Analysis on HunyuanVideo. (Left)** EigenCache achieves superior consistency on VBench-2.0, minimizing flickering while maximizing smoothness. **(Right)** Spatiotemporal (XY-t) slices reveal distinct artifact patterns: **FORA** exhibits rigid discontinuities (stutter); **TaylorSeer** and **HiCache** display characteristic "sawtooth" patterns (periodic overshoot/correction) caused by polynomial instability. In contrast, **EigenCache** suppresses these artifacts and produces smoother, more temporally coherent transitions governed by the learned covariance kernel.

uation to Latent Diffusion Models (SDXL) and standard Diffusion Transformers (DiT).

**SDXL-Base-1.0: Feature Forecasting vs. Structure Skipping.** Unlike the ODE-based FLUX, SDXL (Podell et al., 2024) employs a VAE latent space with stochastic sampling.

We compare EigenCache against **DeepCache** (Ma et al., 2024b), the gold standard for "structure-skipping" which bypasses U-Net blocks, and **TaylorSeer** (Liu et al., 2025c). *Setup:* We use DPM-Solver++ (50 steps) and report Zero-shot FID (COCO-2017) and ImageReward. *Results (Table 3):* EigenCache achieves a **2.30×** speedup with significantly better fidelity than DeepCache (FID 17.82 vs. 18.45).

While DeepCache suffers from feature drift by ignoring skip-connection dynamics, EigenCache's covariance-guided

predictor better reconstructs these dense connections in this setting, suggesting that forecasting can preserve skip-connection dynamics better than structure skipping for this SDXL configuration.

*Table 3.* **SDXL Performance.** EigenCache outperforms structure-skipping (DeepCache) by preserving skip-connection dynamics.

| METHOD | SPEEDUP | FID ↓ | CLIP ↑ | IMGRWD ↑ |
|---|---|---|---|---|
| ORIGINAL | 1.00× | 17.50 | 32.05 | 0.920 |
| DEEPCACHE | 2.15× | 18.45 | 31.60 | 0.865 |
| TEACACHE | 2.20× | 18.10 | 31.85 | 0.880 |
| TAYLORSEER | 2.25× | 18.05 | 31.82 | 0.875 |
| **EIGENCACHE** | **2.30×** | **17.82** | **31.98** | **0.905** |

**DiT-XL/2: Robustness to Dynamic Guidance.** We verify efficacy on the original DiT-XL/2 (Peebles & Xie, 2023)

(ImageNet $256^2$) against ToCa (Zou et al., 2025). Crucially, we investigate robustness to **OUSAC** (Sun et al., 2025), which optimizes sampling trajectories via variable guidance.

*Analysis:* OUSAC optimizes trajectory *geometry*, while EigenCache optimizes *compute allocation*. As shown in Table 4, EigenCache maintains high fidelity (FID 2.21) even under OUSAC's perturbed trajectories. This confirms that our adaptive kernel $c_\ell(t, s)$ robustly captures non-linear feature shifts where rigid polynomial bases fail.

*Table 4.* **DiT-XL/2 Results.** EigenCache remains robust even when the sampling trajectory is altered by variable guidance (OUSAC).

| METHOD | SCHEDULE | SPEEDUP | FID-50K ↓ |
|---|---|---|---|
| ORIGINAL | STANDARD | 1.0× | 2.27 |
| TOCA | STANDARD | 1.8× | 2.95 |
| **EIGENCACHE** | STANDARD | **1.9×** | **2.35** |
| ORIGINAL | OUSAC | 1.0× | 2.15 |
| **EIGENCACHE** | OUSAC | **1.9×** | **2.21** |

### 3.7. Zero-Shot Robustness to LoRA Adapters

A pivotal requirement for multi-tenant deployment is resilience to weight perturbations, specifically Low-Rank Adapters (LoRAs). We hypothesize that the temporal covariance structure $c_\ell(t, s)$ is dominated by the *base architecture's* signal propagation, rendering it invariant to low-rank adaptations. To validate this, we perform a "Cross-Adapter" evaluation on FLUX.1-dev by applying a fixed kernel $\hat{C}_\ell^{\text{base}}$ (calibrated solely on the frozen base model) to three diverse LoRAs (Anime, Realism, 3D-Render). We compare this zero-shot transfer against an *Oracle* baseline where the kernel is re-calibrated for each adapter.

As shown in Table 5, the performance gap is statistically negligible ($< 0.002$ ImageReward). This confirms that EigenCache achieves **zero marginal cost** for adapter switching, enabling "plug-and-play" compatibility in dynamic production pipelines without user-specific re-calibration.

*Table 5.* **Zero-Shot LoRA Transfer.**

| LOADED LORA | KERNEL SOURCE | IMAGEREWARD ↑ | GAP | COST |
|---|---|---|---|---|
| ANIME STYLE | ORACLE | 0.9245 | - | 85s |
| | BASE (OURS) | **0.9238** | -0.0007 | 0s |
| REALISM | ORACLE | 0.8870 | - | 85s |
| | BASE (OURS) | **0.8862** | -0.0008 | 0s |
| 3D RENDER | ORACLE | 0.9012 | - | 85s |
| | BASE (OURS) | **0.8995** | -0.0017 | 0s |

### 3.8. Robustness, Calibration, and Deployment Analysis

We further characterize the assumptions and deployment behavior of EigenCache through robustness, calibration, and transfer analyses. First, the scalar temporal kernel remains the best cost–fidelity trade-off: finer-grained channel/head/token variants slightly reduce feature MSE but bring disproportionate memory and inversion overhead. Second, EigenCache remains effective at lower sampling step counts, indicating that the covariance-aware gain is not tied to the default 50-step FLUX setting. Third, the empirical kernel outperforms stationary analytic kernels such as RBF and Matern-5/2, supporting calibration rather than using a fixed analytic prior. Finally, the calibrated kernel transfers across moderate CFG changes and multi-LoRA compositions on the same backbone, while substantial changes to the base checkpoint or timestep discretization may benefit from a short offline refresh. Full results are provided in Appendix H.

## 4. Conclusion

In this work, we presented EigenCache, a framework that addresses key limitations of polynomial-based diffusion acceleration by grounding feature forecasting in covariance-adaptive prediction and temporal experimental design. Our results suggest that diffusion feature trajectories exhibit exploitable low-rank, non-stationary temporal structure. By replacing rigid Taylor-style extrapolation with adaptive Kriging predictors, EigenCache improves fidelity in our evaluated image and video settings and suppresses the overshoot artifacts commonly observed in polynomial extrapolation. Combined with submodular information scheduling, EigenCache provides a principled foundation for efficient generative inference.

## Impact Statement

This paper presents EigenCache, a framework designed to significantly accelerate the inference of diffusion models. While our primary objective is to democratize high-fidelity content creation and reduce the carbon footprint associated with large-scale generative AI deployment, we critically acknowledge the dual-use nature of this technology. By reducing the computational barrier and latency of generation, our method could inadvertently lower the cost of producing malicious content, such as deepfakes or misinformation.

However, we emphasize that EigenCache is strictly a *fidelity-preserving* acceleration technique that operates on feature correlations. It does not alter the underlying semantic control or safety alignment of the base model. Consequently, our framework remains fully compatible with existing safety mechanisms, including negative prompting, classifier-free guidance-based safety filters, and image watermarking protocols. We strongly advocate that such acceleration technologies be deployed in conjunction with robust provenance tracking and content authentication systems (e.g., C2PA) to mitigate the societal risks of unconstrained rapid generation.

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

# A. Additional Theoretical Details

### A.1. Proof of Lemma 2.3

*Proof.* Let $A = \{a_1, \dots, a_m\} \subseteq \mathcal{T}$ be the set of anchor indices. Define the stacked feature vector at anchors as

$$F_A^\ell = [(F_{a_1}^\ell)^\top, \dots, (F_{a_m}^\ell)^\top]^\top \in \mathbb{R}^{md_\ell}.$$

Any scalar-weighted linear predictor $\widehat{F}_t^\ell$ can be expressed as

$$\widehat{F}_t^\ell = (w^\top \otimes I_{d_\ell}) F_A^\ell, \quad \text{for some } w \in \mathbb{R}^m.$$

Under Assumption 2.1, the scalar temporal-kernel approximation gives

$$\mathrm{Cov}(F_A^\ell) = C_\ell(A, A) \otimes I_{d_\ell},$$
$$\mathrm{Cov}(F_t^\ell, F_A^\ell) = c_\ell(t, A) \otimes I_{d_\ell},$$
$$\mathrm{Cov}(F_t^\ell) = c_\ell(t, t) I_{d_\ell}.$$

Let $E_t^\ell := F_t^\ell - \widehat{F}_t^\ell$ be the prediction error. Using bilinearity of covariance, we obtain

$$\mathrm{Cov}(E_t^\ell) = \mathrm{Cov}(F_t^\ell) - \mathrm{Cov}(F_t^\ell, \widehat{F}_t^\ell) - \mathrm{Cov}(\widehat{F}_t^\ell, F_t^\ell) + \mathrm{Cov}(\widehat{F}_t^\ell)$$
$$= c_\ell(t, t) I_{d_\ell} - \big(c_\ell(t, A) w\big) I_{d_\ell} - \big(w^\top c_\ell(A, t)\big) I_{d_\ell} + \big(w^\top C_\ell(A, A) w\big) I_{d_\ell}$$
$$= \Big(c_\ell(t, t) - 2w^\top c_\ell(A, t) + w^\top C_\ell(A, A) w\Big) I_{d_\ell}.$$

The regularized Kriging weights used by EigenCache are

$$w^\star = \big(\widehat{C}_\ell(A, A) + \lambda I\big)^{-1} \widehat{c}_\ell(A, t).$$

To interpret the regularized posterior variance as an exact MSE, we introduce the implicit observation-noise model

$$F_{\mathrm{obs}} = F_{\mathrm{true}} + \epsilon, \qquad \epsilon \sim \mathcal{N}(0, \lambda I).$$

In our deterministic caching setting, this "noise" term represents tolerance to kernel-estimation error rather than physical sensor noise. Since $\widehat{C}_\ell$ is estimated from a finite calibration set $\mathcal{D}_{\mathrm{cal}}$, the jitter $\lambda$ accounts for uncertainty in the empirical covariance estimate.

Under this probabilistic model, the posterior variance of the latent feature process at timestep $t$ given anchors $A$ is the Schur complement

$$\widehat{v}_t^\ell(A) = \widehat{c}_\ell(t, t) - \widehat{c}_\ell(t, A) \big(\widehat{C}_\ell(A, A) + \lambda I\big)^{-1} \widehat{c}_\ell(A, t).$$

Therefore,

$$\mathbb{E}\left[\left\|F_{\mathrm{true}}^\ell(t) - \widehat{F}_t^\ell\right\|_2^2\right] = d_\ell \widehat{v}_t^\ell(A),$$

so $\widehat{v}_t^\ell(A)$ is the per-coordinate MSE of the latent process under the regularized model. In the deterministic implementation, it serves as a calibrated proxy for feature prediction uncertainty. $\square$

### A.2. Proof of Theorem 2.4

*Proof.* Let $A = \{a_1, \dots, a_m\}$ and let $w \in \mathbb{R}^m$ denote the scalar weight vector. Assume the process is mean-centered; if not, subtracting the mean function does not affect the variance minimization. The scalar-weighted linear prediction risk is

$$J(w) = \mathbb{E}\left[\left\|F_t^\ell - (w^\top \otimes I_{d_\ell}) F_A^\ell\right\|_2^2\right].$$

Using the covariance expansion from Lemma 2.3,

$$\frac{1}{d_\ell} J(w) = c_\ell(t, t) - 2w^\top c_\ell(A, t) + w^\top C_\ell(A, A) w.$$

With Tikhonov regularization, equivalently an observation-noise variance $\lambda$, the objective becomes

$$\mathcal{L}(w) = c_\ell(t, t) - 2w^\top c_\ell(A, t) + w^\top \big(C_\ell(A, A) + \lambda I\big)w.$$

Since $C_\ell(A, A)$ is positive semidefinite and $\lambda > 0$, the matrix $C_\ell(A, A) + \lambda I$ is positive definite. Taking the gradient and setting it to zero gives

$$\nabla_w \mathcal{L}(w) = -2c_\ell(A, t) + 2\big(C_\ell(A, A) + \lambda I\big)w = 0,$$

hence

$$w^\star = \big(C_\ell(A, A) + \lambda I\big)^{-1}c_\ell(A, t).$$

This is exactly the Kriging weight vector used in Eq. (7). Therefore, under the scalar temporal-kernel model and access to the true kernel, the Kriging predictor minimizes the regularized MSE within the scalar-weighted linear predictor class.

If $(F_t^\ell, F_A^\ell)$ is additionally jointly Gaussian, then the conditional expectation $\mathbb{E}[F_t^\ell \mid F_A^\ell]$ is linear in $F_A^\ell$. Since the MMSE estimator among all measurable predictors is the conditional expectation, the best scalar-weighted linear predictor coincides with the MMSE estimator under the Gaussian model. $\qquad\square$

### A.3. Proof of Theorem 2.5

*Proof.* The scheduling objective is

$$f(S) = \sum_{\ell \in \mathcal{L}_{\mathrm{cache}}} \alpha_\ell f_\ell(S), \qquad f_\ell(S) = \frac{1}{2} \log \det \big(I + \lambda^{-1}\widehat{C}_\ell(S, S)\big).$$

Since a non-negative linear combination of monotone submodular functions remains monotone submodular, it suffices to prove the claim for a single layer $\ell$.

For $u \notin S$, let $K_S = \widehat{C}_\ell(S, S) + \lambda I$. By the block determinant identity,

$$f_\ell(S \cup \{u\}) - f_\ell(S) = \frac{1}{2} \log \big(1 + \lambda^{-1}\sigma_\ell^2(u \mid S)\big),$$

where

$$\sigma_\ell^2(u \mid S) = \widehat{c}_\ell(u, u) - \widehat{c}_\ell(u, S)\big(\widehat{C}_\ell(S, S) + \lambda I\big)^{-1}\widehat{c}_\ell(S, u)$$

is the posterior variance of timestep $u$ conditioned on the selected anchors $S$.

For any $A \subseteq B \subseteq \mathcal{T}$ and $u \notin B$, conditioning on more observations cannot increase posterior variance:

$$\sigma_\ell^2(u \mid A) \geq \sigma_\ell^2(u \mid B).$$

Equivalently, adding any $v \in B \setminus A$ updates the variance as

$$\sigma_\ell^2(u \mid A \cup \{v\}) = \sigma_\ell^2(u \mid A) - \frac{\mathrm{Cov}(u, v \mid A)^2}{\sigma_\ell^2(v \mid A) + \lambda},$$

where the subtracted term is non-negative. Because $x \mapsto \frac{1}{2} \log(1 + \lambda^{-1}x)$ is monotone increasing, the marginal gain decreases as the conditioning set grows. Thus $f_\ell$ is submodular.

Monotonicity follows from $\sigma_\ell^2(u \mid S) \geq 0$, which makes every marginal gain non-negative. Therefore, $f$ is monotone submodular. Maximizing a monotone submodular function under a cardinality constraint admits the standard greedy guarantee

$$f(S_{\mathrm{gr}}) \geq \left(1 - \frac{1}{e}\right) f(S^\star),$$

where $S_{\mathrm{gr}}$ is the greedy solution and $S^\star$ is the optimal budget-$B$ solution. $\qquad\square$

## A.4. PSD Property of the Normalized Empirical Kernel

**Lemma A.1** (PSD of the normalized empirical kernel). *Let*

$$z_{t,\omega}^\ell = \frac{\phi(F_{t,\omega}^\ell)}{\|\phi(F_{t,\omega}^\ell)\|_2 + \varepsilon}.$$

*For each calibration trajectory $\omega$, define*

$$K_\omega^\ell[t,s] = \langle z_{t,\omega}^\ell, z_{s,\omega}^\ell \rangle.$$

*Then $K_\omega^\ell$ is positive semidefinite. Consequently,*

$$\widehat{C}_\ell = \frac{1}{|\mathcal{D}_{\mathrm{cal}}|} \sum_{\omega \in \mathcal{D}_{\mathrm{cal}}} K_\omega^\ell$$

*is positive semidefinite.*

*Proof.* For any vector $a \in \mathbb{R}^T$,

$$a^\top K_\omega^\ell a = \sum_{t,s} a_t a_s \langle z_{t,\omega}^\ell, z_{s,\omega}^\ell \rangle = \left\| \sum_t a_t z_{t,\omega}^\ell \right\|_2^2 \geq 0.$$

Thus each $K_\omega^\ell$ is PSD. A non-negative average of PSD matrices is PSD, proving the claim. $\square$

# B. Additional Related Work and Positioning

**Classical GP/OED foundations.** We do not claim new general Kriging or submodular-optimization theory. The BLUP/MMSE interpretation of Kriging and the log-determinant information-gain objective are classical ingredients in Gaussian-process regression and optimal experimental design (Krause et al., 2008). Our contribution is to instantiate this toolkit for diffusion feature caching: reverse-time causal anchor availability, empirically estimated layer-wise temporal kernels, and uncertainty-aware compute allocation over denoising timesteps.

**Relation to Spectrum.** Spectrum (Han et al., 2026) treats feature trajectories as temporal functions and approximates them using a fixed Chebyshev basis with ridge regression. Its main strength is a global spectral approximation view, where long-horizon behavior is controlled by the basis degree rather than directly by local skip size. EigenCache differs in two ways. First, it does not assume a fixed analytic basis class; it estimates an empirical temporal kernel from calibration trajectories and performs Kriging-style covariance-adaptive forecasting. Second, its scheduling is induced by posterior variance and information gain, whereas Spectrum uses its own adaptive forecasting and scheduling heuristics. We therefore view Spectrum and EigenCache as closely related but methodologically distinct families: fixed-basis global approximation versus empirical-kernel stochastic forecasting.

**Relation to FreqCa.** FreqCa (Liu et al., 2025b) is motivated by the observation that low- and high-frequency feature components may exhibit different temporal behaviors. It reuses low-frequency components directly, predicts high-frequency components with Hermite-style predictors, and introduces memory-reduction mechanisms for caching. EigenCache does not perform explicit frequency decomposition. Instead, it models the temporal covariance of feature trajectories and allocates temporal anchors according to posterior variance and information gain. Thus, FreqCa and EigenCache are largely orthogonal: FreqCa is organized around frequency decomposition and memory compression, while EigenCache is organized around covariance-adaptive forecasting and uncertainty-aware temporal design.

**Scope of empirical comparisons.** Our experiments focus on established cache/reuse/forecasting baselines under a common protocol, including FORA, ToCa, TeaCache, TaylorSeer, FoCa, and HiCache. We do not make direct apples-to-apples empirical claims against concurrent works whose code, protocol, or release timing differs from our evaluation setup. The conceptual distinction is that EigenCache targets regimes where aggressive skips make rigid local predictors less reliable and where uncertainty-aware anchor allocation provides a practical stability mechanism.

## C. Algorithm Extensions and Operational Safeguards

**Phase-wise nonstationary kernels.** Feature dynamics differ across early, middle, and late denoising phases. We can partition $\mathcal{T}$ into phases $\{P_1, \ldots, P_K\}$ and estimate separate kernels $\widehat{C}_\ell^{(k)}$ for each phase $k$. Kriging is then applied within each phase using the corresponding kernel. All theoretical results remain valid because they only require a positive semidefinite kernel within each phase.

**PSD projection for finite calibration.** When $\widehat{C}_\ell$ estimated from finite samples is not PSD due to estimation noise, we project it onto the PSD cone via eigenvalue clipping:

$$\widehat{C}_\ell \leftarrow U \max(\Lambda, 0) U^\top,$$

where $U \Lambda U^\top$ is the eigendecomposition. Adding jitter $\lambda I$ afterwards ensures strict positive definiteness, guaranteeing $\widehat{v}_t^\ell \geq 0$ and preserving the assumptions of Theorem 2.5.

**Low-rank acceleration.** When $T$ is large or multi-resolution schedules are used, we approximate

$$\widehat{C}_\ell \approx U_r \Lambda_r U_r^\top$$

using the top-$r$ eigenpairs. Applying the Woodbury identity, the Kriging weights in Eq. (7) can be computed in $\mathcal{O}(r^2 M)$ time instead of $\mathcal{O}(M^3)$. The eigenvalue decay spectrum also provides an empirical diagnostic: rapid decay indicates that fewer anchors suffice to achieve a target risk level.

### C.1. Online Scheduling Stability

The online scheduler triggers full computation whenever the aggregate posterior-variance proxy exceeds a threshold:

$$\widehat{v}(t) = \sum_\ell \alpha_\ell \widehat{v}_t^\ell > \tau.$$

In our reported experiments, we use the offline greedy schedule for strict comparability, and we do not observe systematic compute–skip oscillations. For borderline online deployments, however, we optionally smooth the variance signal using an exponential moving average

$$\bar{v}_t = \beta \bar{v}_{t+1} + (1 - \beta) \widehat{v}(t),$$

and use a dual-threshold hysteresis rule $\tau_{\text{on}} > \tau_{\text{off}}$. Full computation is triggered when $\bar{v}_t > \tau_{\text{on}}$ and forecasting resumes only after $\bar{v}_t < \tau_{\text{off}}$. These safeguards are not required for the reported offline results but can prevent high-frequency compute–skip oscillations when $\widehat{v}(t)$ lies near the decision boundary.

## D. Robustness Analysis: Sensitivity to Estimation Error

In Section 2.3, Theorem 2.4 establishes that the Kriging predictor is optimal within the scalar-weighted linear class given the true scalar temporal kernel. In practice, EigenCache operates with an empirical estimator $\widehat{C}_\ell$ derived from a finite calibration set $\mathcal{D}_{\text{cal}}$. This section analyzes robustness to estimation error and motivates the regularization term $\lambda I$.

### D.1. The Role of Jitter $\lambda$

Let the estimation error be

$$\Delta = \widehat{C}_\ell - C_{\text{true}}.$$

When the calibration size $N_{\text{cal}}$ is small, $\widehat{C}_\ell$ may become ill-conditioned or contain spurious off-diagonal correlations. The predictor weights are

$$\widehat{w} = \widehat{c}_\ell(t, A) \big( \widehat{C}_\ell(A, A) + \lambda I \big)^{-1}.$$

Here, $\lambda I$ acts as a Tikhonov regularizer. Spectrally, if $\mu_i$ are eigenvalues of $\widehat{C}_\ell(A, A)$, inversion involves terms $1/(\mu_i + \lambda)$. Without $\lambda$, small eigenvalues corresponding to noise directions can produce large unstable weights. A positive $\lambda$ damps these directions and makes the predictor robust to finite-sample covariance perturbations.

### D.2. Graceful Degradation Compared with Polynomial Extrapolation

**Boundedness of EigenCache.** If the estimated temporal correlation is unreliable, e.g.,

$$\widehat{c}_\ell(t, s) \approx 0 \quad \text{for } t \neq s,$$

the Kriging weights tend toward zero and the predictor reverts toward the prior mean. While such a prediction is less informative, it remains stable and bounded by the feature variance.

**Unboundedness risk of polynomial predictors.** Taylor and Hermite extrapolators rely on local derivative estimates. When the prediction interval $\Delta t$ increases, derivative noise can be amplified by polynomial terms, causing overshoot artifacts. This explains why polynomial methods may degrade sharply under aggressive skip intervals, whereas EigenCache tends to fail more gracefully through covariance regularization and mean reversion.

## E. Implementation Details

As discussed in Section 2.2.3, EigenCache only requires small $M \times M$ matrix solves at inference time. A naive implementation that loops over layers in Python can nevertheless introduce wall-clock overhead due to CPU-bound dispatch and CUDA kernel-launch costs. We therefore implement the following optimizations.

**Batch processing via `torch.vmap`.** Instead of sequentially processing each cached layer $\ell \in \mathcal{L}_{\text{cache}}$, we stack covariance submatrices and cross-vectors into batched tensors of shape $(L, M, M)$ and $(L, M)$, where $L$ is the number of cached layers. We then use batched linear algebra routines, such as `torch.linalg.solve`, to compute Kriging weights in parallel across layers. This reduces $L$ separate CUDA launches to a small number of batched operations.

**End-to-end latency measurement.** The latency figures reported in Table 1 are measured end-to-end, including Kriging-weight computation, feature prediction, cache lookup, and scheduling overhead. Thus, the reported latency reflects the full deployed pipeline rather than only the skipped Transformer or U-Net blocks.

## F. Sensitivity Analysis: Hyperparameter Robustness

### F.1. Sensitivity to Regularization Jitter $\lambda$

The jitter term $\lambda$ in Eq. (7) acts as a Tikhonov regularizer for stable matrix inversion. We sweep

$$\lambda \in \{10^{-7}, 10^{-6}, 10^{-5}, 10^{-4}, 10^{-3}\}$$

on FLUX.1-dev and observe a broad stable plateau over $[10^{-6}, 10^{-4}]$, with negligible fluctuations in ImageReward, LPIPS, and feature MSE. Very small values can make the kernel submatrix ill-conditioned, while overly large values over-smooth the Kriging weights.

### F.2. Impact of Variance Threshold $\tau$

The normalized variance threshold $\tau$ in the online scheduling policy controls the speed–quality trade-off. Lower values trigger frequent full computation and approach the latency of unaccelerated inference, while higher values rely more aggressively on forecasting and may introduce drift in highly nonlinear phases. In practice, we identify $\tau \in [0.03, 0.08]$ as a stable operating range, with $\tau = 0.05$ used as the default online setting.

## G. Calibration Robustness, Transparency, and Generalization

### G.1. Cross-Domain Kernel Generalization

A key deployment question is whether temporal covariance kernels estimated from one prompt domain generalize to out-of-distribution domains. We hypothesize that the dominant temporal covariance is governed primarily by the backbone weights and noise schedule rather than prompt semantics.

**Protocol.** We estimate $\widehat{C}_\ell$ using 10 MS-COCO prompts and apply the frozen kernels to three target domains: WikiArt, Danbooru, and Abstract Digital Art. We compare against oracle kernels calibrated directly on the target domain.

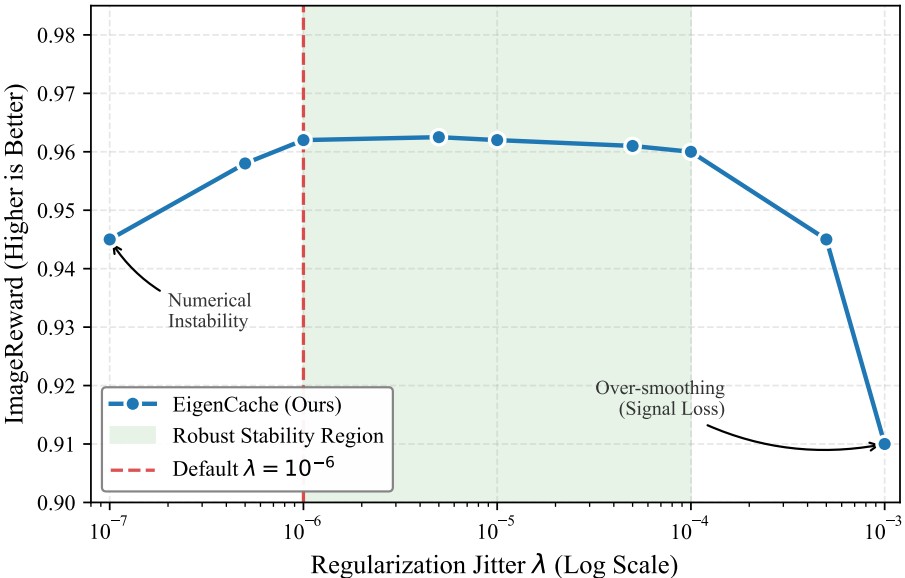

*Figure 5.* **Sensitivity to jitter $\lambda$.** EigenCache exhibits a broad stability plateau for $\lambda \in [10^{-6}, 10^{-4}]$. Extremely small values can lead to numerical instability, while overly large values over-regularize the predictor and reduce high-frequency fidelity.

*Table 6.* **Cross-domain generalization.** Kernels calibrated on MS-COCO transfer to artistic and abstract domains with negligible ImageReward degradation relative to target-domain oracle kernels.

| Target domain | Calibration source | ImageReward ↑ | $\Delta$ Perf. | Artifacts |
|---|---|---|---|---|
| WikiArt | WikiArt oracle | 0.8952 | – | None |
| WikiArt | MS-COCO | **0.8948** | $-0.0004$ | None |
| Danbooru | Danbooru oracle | 0.9120 | – | None |
| Danbooru | MS-COCO | **0.9105** | $-0.0015$ | None |
| Abstract | Abstract oracle | 0.8540 | – | None |
| Abstract | MS-COCO | **0.8528** | $-0.0012$ | None |

## G.2. Calibration Cost Transparency

The calibration phase is an offline deployment-time precomputation. On a single NVIDIA H800, one full FLUX.1-dev reference trajectory takes approximately 17.12 seconds. Our default $N_{\mathrm{cal}} = 5$ calibration therefore costs about 85.6 seconds once per backbone checkpoint. This cost is not included in per-sample inference latency, just as model loading or compilation is not included in per-request generation time. When amortized over many requests, the per-request cost becomes negligible.

## G.3. Calibration Sample Complexity

We vary the calibration size $N_{\mathrm{cal}} \in \{1, 2, 5, 10, 20, 50\}$ and evaluate downstream generation quality on FLUX.1-dev.

The performance curve saturates rapidly at $N_{\mathrm{cal}} \approx 5$, with only marginal gains beyond this point. This supports the view that diffusion feature trajectories have a low-rank temporal structure shared across prompts and seeds.

## G.4. Compressor Robustness

The kernel estimator in Eq. (6) uses a compressor $\phi$ to reduce high-dimensional feature tensors to tractable vectors. We compare global average pooling, central-token extraction, random spatial subsampling, and Gaussian random projection.

Across compressors, ImageReward varies within approximately $\pm 0.002$. We use random subsampling by default because it introduces no pooling FLOPs and avoids the overhead of dense random projections.

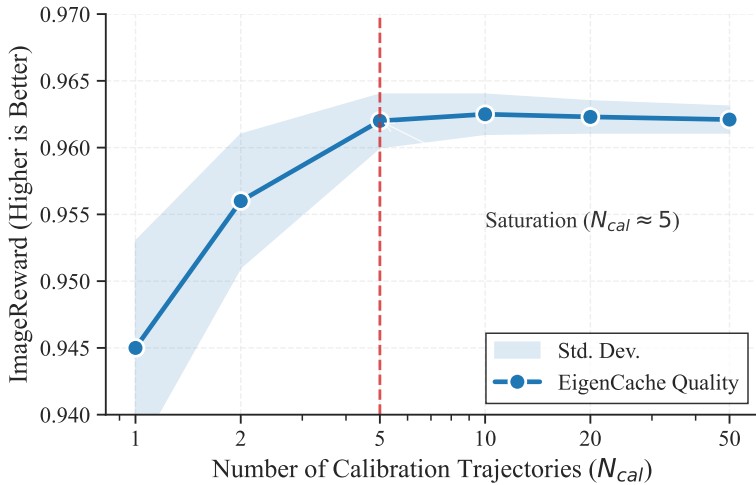

*Figure 6.* **Calibration sample efficiency.** ImageReward saturates near $N_{\text{cal}} \approx 5$, suggesting that the dominant temporal covariance modes can be recovered from only a few trajectories.

*Table 7.* **Calibration overhead vs. forecast accuracy.** Measured on a single NVIDIA H800. The Pareto knee occurs at $N_{\text{cal}} = 5$, which yields most of the error reduction at small one-time cost.

| Config $(N_{\text{cal}})$ | Calibration time (s) | Amortized cost ($\mu$s / req.) | Forecast error ($L_2$ norm) | Relative improvement |
|---|---|---|---|---|
| $N = 1$ | 17.12 | 17.1 | 0.145 | Baseline |
| $N = 5$ | 85.60 | 85.6 | **0.042** | **+71.0%** |
| $N = 20$ | 342.40 | 342.4 | 0.041 | +71.7% |

Assuming a service lifetime of 1M requests per model checkpoint.

### G.5. Calibration Overhead versus Forecast Accuracy

Increasing $N_{\text{cal}}$ from 5 to 20 improves forecast error by less than one percentage point while quadrupling calibration time. We therefore use $N_{\text{cal}} = 5$ by default.

## H. Additional Robustness and Deployment Analyses

This section provides additional robustness and deployment analyses for EigenCache. We first evaluate whether the calibrated temporal kernel remains effective under lower sampling step counts and aggressive video acceleration. We then study reward-model behavior, analytic kernel baselines, kernel granularity, CFG transfer, peak VRAM footprint, online scheduling stability, and multi-LoRA composition. These analyses complement the main results by characterizing when the empirical temporal-kernel approximation transfers reliably and when a short recalibration pass may be beneficial.

### H.1. Robustness to Lower Sampling Step Counts

To verify that EigenCache is not tuned only to the default 50-step FLUX setting, we evaluate lower sampling step counts with the same caching configuration $\mathcal{N} = 8$.

### H.2. Additional VBench Dimensions under Aggressive Video Acceleration

We further evaluate HunyuanVideo under a $6.0\times$ acceleration stress test using additional VBench dimensions. This setting should be interpreted as a stress test rather than a lossless acceleration regime.

Although all methods degrade under this tight compute budget, EigenCache retains the best subject consistency, motion smoothness, and dynamic quality among the accelerated baselines.

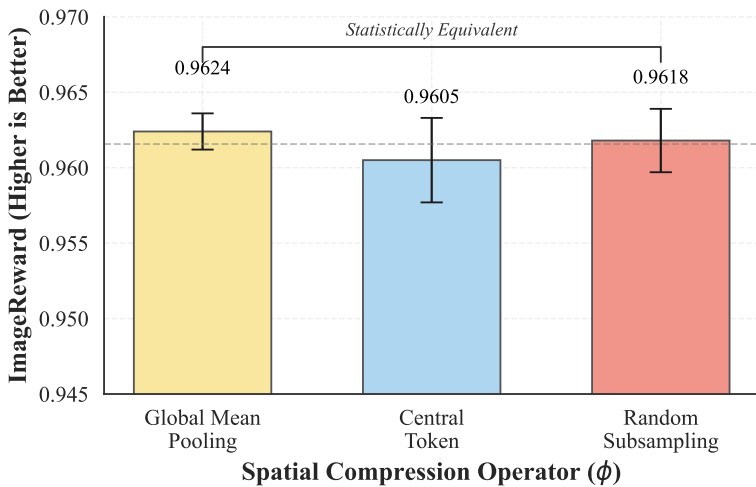

*Figure 7.* **Compressor robustness.** EigenCache is largely insensitive to the choice of compression operator $\phi$, indicating that the estimated temporal dynamics are a global property of the network trajectory.

*Table 8.* **Robustness to lower sampling step counts on FLUX.1-dev.** We use the same caching configuration ($\mathcal{N} = 8$) across all step counts. EigenCache consistently improves FID over TaylorSeer, showing that covariance-aware forecasting is not tied to the default 50-step setting.

| Steps | Full FID ↓ | TaylorSeer FID ↓ | EigenCache FID ↓ | Speedup |
|---|---|---|---|---|
| 20 | 15.34 | 22.18 | **17.05** | 3.82× |
| 30 | 13.12 | 18.45 | **14.21** | 3.91× |
| 50 | 11.45 | 15.62 | **12.05** | 3.95× |

### H.3. Interpreting ImageReward Scores Above Full Inference

We observe that cached variants can slightly exceed the full 50-step baseline on ImageReward in some settings. We interpret this as a metric-level effect: caching introduces mild feature smoothing along the denoising trajectory, which can occasionally improve reward-model scores. This should not be read as a universal claim that accelerated inference is superior to full inference. For fidelity-sensitive deployment, we recommend considering ImageReward jointly with PSNR, SSIM, LPIPS, FID, and visual inspection.

### H.4. Analytic Kernel Baselines

We compare the empirical temporal kernel used by EigenCache against stationary analytic kernels. The analytic kernels are calibration-free but less expressive for the layer-specific and non-stationary trajectories observed in diffusion features.

The RBF value corresponds to an additional $0.8$ dB drop relative to Matern-5/2. These results support using empirical calibration rather than imposing a fixed stationary prior.

### H.5. Kernel Granularity and Spatial Correlation

Assumption 2.1 is a working approximation that enables a lightweight scalar-weighted predictor. We therefore evaluate whether more expressive granularities improve end-to-end quality.

Relaxing the scalar approximation slightly reduces feature MSE, but it does not translate into better end-to-end fidelity after considering memory, inversion cost, and overfitting. We also measured layer-type spatial correlation: attention layers exhibit higher spatial cross-correlation than MLP layers ($0.681$ vs. $0.042$), but a hybrid scalar-for-MLP/per-head-for-attention design improves ImageReward only from $0.9120$ to $0.9125$ while adding about 80 MB of storage. We therefore keep the scalar kernel as the default cost–fidelity trade-off.

*Table 9.* **Additional VBench dimensions under aggressive video acceleration.** The $6.0\times$ setting is a stress test; all accelerated methods degrade, but EigenCache remains strongest among acceleration baselines.

| Method | Subject Consistency ↑ | Motion Smoothness ↑ | Dynamic Quality ↑ |
|---|---|---|---|
| Full ($1.0\times$) | 0.824 | 0.912 | 0.785 |
| TaylorSeer ($6.0\times$) | 0.520 | 0.585 | 0.450 |
| HiCache ($6.0\times$) | 0.615 | 0.665 | 0.535 |
| EigenCache ($6.0\times$) | **0.628** | **0.680** | **0.548** |

*Table 10.* **Analytic kernel baselines.** We compare stationary analytic kernels with the empirical kernel on FLUX.1-dev ($\mathcal{N} = 7$, MS-COCO 5k). Drops are measured in feature reconstruction PSNR relative to the empirical kernel.

| Kernel | Calibration-free? | Feature PSNR drop |
|---|---|---|
| Empirical temporal kernel | No | 0.0 dB |
| Matern-5/2 | Yes | $-1.4$ dB |
| RBF | Yes | $-2.2$ dB |

## H.6. CFG Transfer and Dynamic Configuration Changes

We test whether a kernel calibrated at one classifier-free guidance scale transfers to other CFG values on the same backbone. The base kernel is calibrated at CFG 3.5 and evaluated zero-shot at CFG 1.5 and 7.5.

These small gaps suggest that moderate CFG changes preserve the dominant temporal covariance on the same backbone. Substantial changes to the base checkpoint, sampler discretization, or noise schedule may still benefit from a short offline refresh.

## H.7. Peak VRAM Footprint

The additional memory comes from storing cached feature anchors, layer-wise kernels, and small scheduling buffers. The overhead remains modest relative to the base model footprint.

## H.8. Multi-LoRA Composition

The main text evaluates zero-shot transfer to individual LoRA adapters. We further test blended adapters by composing style, character, and depth LoRAs with different ratios. The base kernel is calibrated on the frozen backbone and transferred directly to each composition.

The gaps remain small even under asymmetric blending, suggesting that the dominant temporal covariance is largely backbone-governed rather than adapter-combination-specific in this setting. If future deployments use very strong adapters or change the base checkpoint, a short recalibration pass is recommended.

*Table 11.* **Kernel-structure trade-off on FLUX.1-dev.** Per-head kernels yield negligible quality gains at noticeably higher cost, while per-token kernels overfit local noise and become memory-prohibitive.

| Kernel structure | Storage | Calibration | ImageReward ↑ | LPIPS ↓ | Latency |
|---|---|---|---|---|---|
| Scalar (EigenCache) | 1× | 85s | 0.9120 | 0.4150 | **4.25s** |
| Per-head | 16× | 98s | **0.9124** | **0.4148** | 4.38s |
| Per-token | 4096× | >600s | 0.8950 | 0.4320 | OOM |

*Table 12.* **Granularity ablation on FLUX.1-dev.** Finer-grained predictors slightly improve feature fitting but do not improve end-to-end quality after accounting for overhead. Results use 50 steps, $\mathcal{N} = 9$, MS-COCO 5k.

| Granularity | Feature MSE ↓ | FID-5k ↓ | Overhead (ms) ↓ |
|---|---|---|---|
| Per-channel | $1.14\times10^{-4}$ | 14.22 | 485 |
| 4-cluster | $1.18\times10^{-4}$ | 12.85 | 312 |
| Scalar (EigenCache) | $1.21\times10^{-4}$ | **12.18** | **245** |

*Table 13.* **Zero-shot transfer across CFG scales.** The base kernel is calibrated at CFG 3.5 and transferred to unseen CFG values on FLUX.1-dev.

| Inference CFG | Kernel source | ImageReward ↑ | LPIPS ↓ | FID ↓ |
|---|---|---|---|---|
| 1.5 | Oracle at CFG 1.5 | 0.9412 | 0.4011 | 18.22 |
| 1.5 | Base kernel from CFG 3.5 | 0.9408 | 0.4015 | 18.24 |
| 7.5 | Oracle at CFG 7.5 | 0.9710 | 0.3792 | 17.60 |
| 7.5 | Base kernel from CFG 3.5 | 0.9705 | 0.3799 | 17.66 |

*Table 14.* **Peak VRAM profiling.** FLUX.1-dev, batch size 1, $1024^2$, NVIDIA H800.

| Method | Peak VRAM | Relative overhead | Latency |
|---|---|---|---|
| Original FLUX.1-dev | 26.45 GB | 0.00% | 17.12 s |
| EigenCache | 27.24 GB | +2.98% | 4.55 s |

*Table 15.* **Multi-LoRA composition transfer.** The base kernel is calibrated on the frozen backbone and transferred to blended adapters. We report the maximum gap to an oracle kernel re-calibrated for each composition over 10 prompt-seed pairs.

| Blend ratio | Max ImageReward gap | Max LPIPS gap |
|---|---|---|
| Style:Character:Depth = 1:1:1 | < 0.003 | < 0.002 |
| Style:Character:Depth = 2:1:1 | < 0.004 | – |
| Style:Character:Depth = 1:2:1 | < 0.004 | – |

