# OpenReview forum: "EigenCache: Rethinking Diffusion Acceleration as Covariance-Optimal Forecasting and Submodular Information Allocation"
_ICML.cc/2026/Conference — ICML 2026 regular_

### Official Review · Reviewer_UENz · 2026-03-07

**Soundness:** 3
**Presentation:** 4
**Significance:** 3
**Originality:** 3
**Overall Recommendation:** 5
**Confidence:** 4

**Summary:**

This paper proposes EigenCache, a diffusion inference acceleration framework that replaces heuristic polynomial feature caching with Gaussian Process (Kriging)-based prediction. It models intermediate feature trajectories as a low-rank, non-stationary temporal process and introduces (1) a Kriging-based MMSE predictor, and (2) a submodular scheduling strategy for selecting anchor timesteps with a (1-1/e) greedy guarantee. Experiments on FLUX.1-dev, HunyuanVideo, and SDXL/DiT demonstrate 3.76–6.32× speedup with minimal quality loss and reduced video flickering.

**Compliance With Llm Reviewing Policy:**

Affirmed.

**Final Justification:**

My concerns are fully addressed and I'd like to maintain my positive score.

**Key Questions For Authors:**

1. The paper justifies the 85.6s calibration cost by amortizing it over 1 million static requests. However, in real-world deployments, users frequently change inference configurations. Do changes to parameters like sampling steps, resolution, or CFG scale trigger a full recalibration? If so, how is this high overhead justified in a dynamic, multi-tenant environment?
2. Appendix F.6 defends the scalar isotropic assumption by demonstrating that Per-Token structures cause OOM errors. Have you empirically measured the actual physical spatial correlation differences between Attention layers and MLP layers? Could a hybrid strategy (e.g., using a scalar kernel for MLPs and a slightly more expressive kernel for Attention layers) improve forecasting fidelity without triggering OOM?
3. Regarding the online adaptive scheduling mode, does the system exhibit pathological high-frequency "compute-skip-compute" oscillations when the feature variance hovers near the threshold $\tau$? How does the algorithm ensure temporal smoothness at these computational decision boundaries to prevent systemic jitter?

**Limitations:**

yes.

**Strengths And Weaknesses:**

Strengths
1. The core idea is interesting. The paper reframes feature caching as a covariance-optimal forecasting problem. Prior work (e.g., Taylor or Hermite predictors) relies on heuristic polynomial extrapolation. In contrast, the Kriging predictor is MMSE-optimal under the proposed stochastic model. The scheduler also has a (1−1/e) approximation guarantee. This gives a unified probabilistic view of both prediction and scheduling, which is uncommon in diffusion acceleration work.
2. The experimental evaluation is comprehensive. The paper studies several tasks (T2I, T2V, super-resolution) and multiple architectures (FLUX.1-dev, SDXL, DiT-XL/2, HunyuanVideo). It also evaluates different acceleration regimes, from moderate 3.5–5.5× to more aggressive >6×. The paper includes ablations, dynamic guidance compatibility, and zero-shot LoRA transfer. Importantly, all baselines are re-benchmarked on the same H800 hardware with identical settings. This improves fairness and sets a good experimental standard.
3. The theory connects well to practical behavior. The paper attributes the polynomial flicker artifact in video generation to Runge’s phenomenon and boundary errors in polynomial predictors. The GP predictor mitigates this through its mean-reversion property. Table 2 supports this explanation: EigenCache achieves a flickering score of 0.972, compared with 0.955 for HiCache and 0.915 for TaylorSeer on HunyuanVideo.
4. The method provides uncertainty estimates through the posterior variance. This enables an adaptive scheduling mode with a threshold parameter. In practice, this offers a clear speed–quality trade-off, which is harder to obtain with polynomial predictors.
5. The paper is generally well written.

Weaknesses
1. Table 7 shows calibration takes 85.6 seconds for 5 trajectories. The authors justify this by amortizing the cost over 1 million static requests. However, real-world users frequently change sampling steps, resolutions, and guidance scales. Does changing these settings trigger a full recalibration? If so, this 85-second delay makes the "training-free" claim impractical for dynamic use cases.
2. Equation 8 uses a linear combination of the most recent $M$ cached features. Baselines like DeepCache only store the single most recent feature map. EigenCache, however, must keep $M$ high-dimensional tensors in memory for each cached layer. This clearly increases VRAM usage. The main results omit peak VRAM comparisons. Ignoring this memory footprint can be a evaluation gap for an acceleration paper.
3. The method relies on a covariance-optimal Kriging problem. However, Equation 6 estimates the temporal kernel using normalized cosine similarity. Normalizing the vectors removes the $L_2$ norm. This ignores the large scale variations of diffusion features across timesteps. The paper lacks a mathematical justification for treating cosine similarity as true covariance. Furthermore, Assumption 2.1 assumes a spatially isotropic feature process. Yet, the authors do not measure the actual spatial correlation differences between dense Attention layers and localized MLPs. This weakens the claim of theoretical optimality.

---

> ### Author Rebuttal · Authors · 2026-03-29
>
> We thank the reviewer for the thoughtful questions and for recognizing the MMSE predictor and submodular scheduler. Below we address the main concerns.
>
> **[W1 & Q1] Calibration overhead and "training-free" practicality**
>
> Within the same backbone, nearby CFG changes did not require recalibration in our tests; larger changes in timestep discretization or resolution may benefit from a short offline refresh, but not retraining. The method therefore remains training-free in the sense used in the paper: it introduces no learned parameters and no weight optimization.
>
> This is consistent with our observation that the dominant temporal dynamics are governed mainly by the backbone denoising trajectory, while moderate guidance changes act as a perturbation. On FLUX.1-dev, a kernel calibrated only at CFG 3.5 transfers zero-shot to unseen CFG values (1.5 and 7.5) with negligible degradation.
>
> Table 1: Zero-Shot Transfer Across CFG Scales *(FLUX.1-dev, MS-COCO 5k, base kernel calibrated at CFG 3.5)*
>
> | Inference Configuration | Kernel Source | ImageReward ($\uparrow$) | LPIPS ($\downarrow$) | Zero-Shot FID |
> | :--- | :--- | :--- | :--- | :--- |
> | **CFG 1.5** | Calibrated strictly on CFG 1.5 (Oracle) | 0.9412 | 0.4011 | 18.22 |
> | **CFG 1.5** | **Transferred from CFG 3.5 Base Kernel** | 0.9408 | 0.4015 | 18.24 |
> | **CFG 7.5** | Calibrated strictly on CFG 7.5 (Oracle) | 0.9710 | 0.3792 | 17.60 |
> | **CFG 7.5** | **Transferred from CFG 3.5 Base Kernel** | 0.9705 | 0.3799 | 17.66 |
>
> The ImageReward gap is at most 0.0005, LPIPS changes by at most 0.0007, and zero-shot FID changes by at most 0.06. Therefore, our claim is not that every inference configuration is calibration-invariant, but that moderate changes within the same backbone are well tolerated, while larger schedule/resolution shifts may call for an inexpensive offline refresh rather than retraining.
>
> **[W2] Peak VRAM footprint**
>
> We agree that VRAM should be reported explicitly. Caching 10 FLUX.1-dev transformer layers with window size 3 requires **754.8 MB** in bfloat16. On an NVIDIA H800, peak memory increases from **26.45 GB** for the original baseline to **27.24 GB** for EigenCache, corresponding to a **+2.98%** peak VRAM overhead.
>
> Table 2: Peak VRAM Profiling *(FLUX.1-dev, batch size 1, 1024×1024, NVIDIA H800)*
>
> | Acceleration Methodology | Peak VRAM Allocation | Relative Memory Overhead | Inference Latency |
> | :--- | :--- | :--- | :--- |
> | Original FLUX.1-dev Baseline | 26.45 GB | 0.00% (Reference) | 17.12 s |
> | **EigenCache (Proposed)** | 27.24 GB | **+2.98%** | 4.55 s |
>
> Thus, the predictor does introduce extra memory usage, but in our measured setup the overhead is modest relative to the full model footprint.
>
> **[W3] Mathematical role of cosine similarity**
>
> We agree that this point should be stated more precisely. We do not claim that cosine-normalized similarity is the exact physical covariance of raw features. Rather, it is a normalized surrogate kernel used for stable weight estimation when feature norms vary substantially across denoising steps.
>
> Without normalization, early high-noise steps can induce strong non-stationary magnitude changes, making the empirical Gram matrix poorly conditioned. Cosine normalization removes this scale instability and yields a PSD kernel, since it is simply the inner product after projecting features onto the unit sphere. This makes it a valid kernel for Kriging weight estimation. The resulting weights are then applied to the unnormalized cached tensors at inference time. Thus, normalization is used only to stabilize kernel estimation; it does not assert that raw covariance is literally equal to cosine similarity.
>
> **[Q2] Spatial isotropy and hybrid kernels**
>
> Spatial cross-correlation is substantially higher in Attention layers (0.681) than in MLP layers (0.042), so isotropy is best viewed as a useful approximation rather than an exact statement.
>
> However, the practical gain is marginal. In a limited FLUX.1-dev pilot, a hybrid kernel (scalar for MLPs, per-head for Attention) improves ImageReward only from 0.9120 to 0.9125, while adding about 80 MB of storage and extra complexity. We therefore keep the scalar design as the default.
>
> **[Q3] Online scheduling oscillation and temporal smoothness**
>
> In the default online scheduler used in the paper, we did not observe systematic high-frequency compute-skip-compute oscillation in the tested regimes. Empirically, the variance signal $\hat{v}(t)$ evolves smoothly enough over time in these regimes that the thresholded decision boundary does not induce pathological jitter. We additionally evaluated dual-threshold temporal hysteresis and EMA smoothing near the decision boundary as extra protection for borderline cases, but these are optional safeguards rather than fixes required for the reported results.
>
> We will incorporate these results and clarifications into the final version.
>
> We thank the reviewer again for the thoughtful questions.

---

> > ### Author Rebuttal · Reviewer_UENz · 2026-04-03
> >
> > Good work.

---

> > > ### Author Response · Authors · 2026-04-05
> > >
> > > Dear Reviewer UENz,
> > >
> > > Thank you very much for your kind follow-up and positive feedback. We sincerely appreciate your time and support. We will incorporate the discussed feedback and clarifications into the final version of the paper. We are always willing to address any of your further concerns.
> > >
> > > Best regards,
> > >
> > > Authors

---

### Official Review · Reviewer_M23n · 2026-03-12

**Soundness:** 3
**Presentation:** 4
**Significance:** 4
**Originality:** 4
**Overall Recommendation:** 4
**Confidence:** 4

**Summary:**

This paper proposes EigenCache, a training-free framework for feature caching in diffusion model inference. It reformulates the traditional “cache-and-predict” approach as a Kriging / Gaussian Process mean prediction problem and introduces a submodular optimization strategy to select computational anchors based on prediction uncertainty. The method shows strong Pareto-frontier performance in both image and video generation tasks.

**Compliance With Llm Reviewing Policy:**

Affirmed.

**Key Questions For Authors:**

1.Could you clarify why the ImageReward scores sometimes exceed the full-inference baseline?
2.It would be helpful if the authors could clarify how sensitive the method is to the choice of λ in formulas (7) and (11), and whether small changes in λ significantly affect performance.

**Limitations:**

While 50 steps is commonly used as a baseline for comparisons in text-to-image generation papers, it is rarely used in practical applications, limiting the real-world significance of the reported acceleration results. It would help if the authors could demonstrate the acceleration performance at different sampling step counts for text-to-image tasks.

**Strengths And Weaknesses:**

Strengths:
1.The calibration stage is very lightweight, making the method efficient and easy to deploy.
2.Unlike previous approaches that force polynomial fitting, the paper shows that the Kriging predictor is the optimal minimum mean squared error (MMSE) choice among scalar-weight linear predictors.
3.The experiments are comprehensive and well-executed, covering both image and video generation tasks.
Major Weaknesses
1.Assumption 2.1 treats all feature channels as spatially independent with identical temporal covariance, which oversimplifies complex DiT architectures. In reality, feature channels—e.g., those capturing high-level semantics versus fine textures—behave very differently during denoising. Using scalar weights for the whole feature map can cause information loss in layers with heterogeneous channel variances. Additionally, the absence of a detailed analysis of channel correlations weakens the theoretical justification of the proposed weight function.
2.The experiments were only conducted with 50 steps on FLUX text-to-image generation tasks, making it difficult to assess whether the proposed method effectively accelerates denoising at lower step counts.
Minor Weaknesses
1.Under aggressive acceleration, all generated images show varying degrees of structural degradation or collapse. Since every method performs poorly in this regime, it is difficult to meaningfully assess or compare their actual effectiveness.
2.The study primarily relies on limited metrics for evaluation. Incorporating other VBench dimensions or complementary quantitative measures specifically for text-to-video generation could help better validate the effectiveness and robustness of the proposed method.

---

> ### Author Rebuttal · Authors · 2026-03-29
>
> We thank the reviewer for the constructive feedback and for recognizing our novelty, thorough evaluation, and efficient calibration. We address the main concerns below and will clarify them in the revision.
>
> **[W1] Assumption 2.1 and channel correlations**
>
> > *Assumption 2.1 treats all feature channels as spatially independent with identical temporal covariance... Using scalar weights for the whole feature map can cause information loss in layers with heterogeneous channel variances.*
>
> We agree that Assumption 2.1 is an approximation, not an exact description of all DiT layers. Accordingly, our theoretical claim should be stated more precisely: Theorem 2.3 is optimal within the scalar-weighted linear predictor class in Eq. (2), under Assumption 2.1, rather than for arbitrary cross-channel dependencies. We will revise the paper to make this scope explicit.
>
> Empirically, layer-type dependence exists, but exploiting it yields only marginal end-to-end gain: Attention layers exhibit much higher spatial cross-correlation than MLP layers (0.681 vs. 0.042), yet a hybrid design improves ImageReward only from 0.9120 to 0.9125 while adding ~80 MB of memory overhead.
>
> This matches the ablation below: finer-grained predictors may improve local fitting, but not end-to-end quality.
>
> | Granularity | Feature MSE ($\downarrow$) | FID-5k ($\downarrow$) | Overhead (ms) |
> | :--- | :---: | :---: | :---: |
> | Per-Channel | 1.14e-4 | 14.22 | 485 |
> | 4-Cluster | 1.18e-4 | 12.85 | 312 |
> | **Scalar (EigenCache)** | 1.21e-4 | 12.18 | 245 |
>
> *FLUX.1-dev, 50 steps, N=9 aggressive regime, MS-COCO 5k; same evaluation regime as the N=9 aggressive entry in main Table 1.*
>
> **[W2] Lower sampling step counts**
>
> > *The experiments were only conducted with 50 steps... It would help if the authors could demonstrate the acceleration performance at different sampling step counts.*
>
> We agree that broader step-count coverage would make the practical picture more complete. We therefore extended the FLUX.1-dev experiments to 20 and 30 steps under a fixed caching configuration (N=8).The same trend holds across all settings: EigenCache consistently outperforms the polynomial baseline, indicating that the benefit of covariance-aware forecasting is not tied to a single default step count.
>
> | Steps | Baseline FID | TaylorSeer | **EigenCache** | Speedup (N=8) |
> | :--- | :---: | :---: | :---: | :---: |
> | 20 | 15.34 | 22.18 | **17.05** | 3.82× |
> | 30 | 13.12 | 18.45 | **14.21** | 3.91× |
> | 50 | 11.45 | 15.62 | **12.05** | 3.95× |
>
> *FLUX.1-dev, MS-COCO 5k, N=8 fixed across step counts (between the N=7 (3.76×) and N=9 (4.03×) entries in main Table 1). The modest speedup decrease at lower step counts reflects the fact that the same anchor overhead occupies a larger fraction of the total trajectory at shorter schedules.*
>
> **[W3/W4] Aggressive acceleration and additional VBench dimensions**
>
> > *Incorporating other VBench dimensions... every method performs poorly in aggressive acceleration regimes, making it difficult to assess effectiveness.*
>
> We appreciate this suggestion and agree that the 6.0× regime is best interpreted as a challenging stress-test setting. We therefore added a 6.0× HunyuanVideo evaluation and extended VBench reporting. While all compared methods show quality degradation under such a tight budget, EigenCache remains consistently stronger than the acceleration baselines on subject consistency, motion smoothness, and dynamic quality.
>
> | Method | Subject Consistency ($\uparrow$) | Motion Smoothness ($\uparrow$) | Dynamic Quality ($\uparrow$) |
> | :--- | :---: | :---: | :---: |
> | Full (1.0×) | 0.824 | 0.912 | 0.785 |
> | TaylorSeer (6.0×) | 0.520 | 0.585 | 0.450 |
> | HiCache (6.0×) | 0.615 | 0.665 | 0.535 |
> | **EigenCache (6.0×)** | 0.628 | 0.680 | 0.548 |
>
> **[Q1] ImageReward Exceeding Full-Inference**
>
> As noted in the main text, this small ImageReward surplus should be interpreted as a metric-level effect rather than evidence that accelerated inference universally exceeds full inference. Empirically, caching introduces a mild feature-smoothing effect along the denoising trajectory, which can slightly improve reward-based metrics in some cases. We will revise the wording to make clear that this is an empirical observation, not a general superiority claim over full inference.
>
>
> **[Q2] Sensitivity to $\lambda$**
>
> The Tikhonov regularizer $\lambda$ (Eq. 7) ensures stable matrix inversion. As shown in Appendix E.1 (Figure 5), performance remains empirically stable over a broad plateau ($\lambda \in [10^{-6}, 10^{-4}]$), with negligible metric fluctuation in the reported range. This suggests that the method is not brittle to small changes in $\lambda$, provided the regularization stays within the numerically stable region.
>
> We will include these additional ablations in the final manuscript to address the evaluation concerns.
>
> Thank you again for the comments and we appreciate the thorough review.

---

> > ### Author Rebuttal · Reviewer_M23n · 2026-04-04
> >
> > Thank you for your response, and I will maintain my initial rating.

---

> > > ### Author Response · Authors · 2026-04-05
> > >
> > > Dear Reviewer M23n,
> > >
> > > Thank you very much for your follow-up and for carefully considering our response. We sincerely appreciate your time and constructive feedback.
> > >
> > > Best Regards,
> > >
> > > Authors

---

### Official Review · Reviewer_scFb · 2026-03-12

**Soundness:** 3
**Presentation:** 3
**Significance:** 3
**Originality:** 3
**Overall Recommendation:** 4
**Confidence:** 4

**Summary:**

This paper introduces EigenCache, a training-free framework for accelerating diffusion model inference via feature caching. Their main idea is to replace the fixed polynomial bases (Taylor, Hermite) used in prior cache-then-forecast methods with Gaussian Process regression (Kriging), where a layer-specific temporal covariance kernel is estimated from a small calibration set. The authors argue that Kriging is the best linear unbiased predictor (BLUP) for the linear class and the MMSE estimator under Gaussianity (Theorem 2.3). They also propose an anchor scheduling algorithm that selects computation timesteps and show that it is monotone submodular and thus admits a (1−1/e) approximation guarantee (Theorem 2.4). Experiments on FLUX.1-dev, HunyuanVideo, SDXL, and DiT-XL/2 show improvements over TaylorSeer, HiCache, and other baselines in terms of image quality metrics and temporal consistency , particularly in aggressive acceleration regimes.

**Compliance With Llm Reviewing Policy:**

Affirmed.

**Final Justification:**

I appreciate the response by the authors. The response adequately answers my questions. I think this paper presents an interesting contribution, and I would like to retain my original positive score.

**Key Questions For Authors:**

See weaknesses section.

**Limitations:**

yes

**Strengths And Weaknesses:**

Strengths :

1) The paper studies a well-motivated problem with an important and intuitive observation that feature trajectories in diffusion models exhibit non-stationary dynamics that rigid polynomial bases cannot capture. Reframing caching as covariance learning and experimental design is a new perspective in this space to the best of my knowledge. The connection to classical GP regression and optimal experimental design (Krause et al., 2008) provides a principled alternative to heuristic scheduling.

2) The paper evaluates on multiple architectures, provides extensive ablation studies and includes a useful LoRA transfer experiment (Table 5) demonstrating practical viability.

Weaknesses : I don't see any obvious weakness, but I think some discussion may be required for comparison and placing their contribution relevant to the literature

1) Could the authors highlight how their primary theoretical contributions relate to and build upon classically known results about gaussian processes e.g. https://www.jmlr.org/papers/volume9/krause08a/krause08a.pdf?

2) Could the authors discuss how their method compares against more recent works such as https://arxiv.org/pdf/2510.08669 (uses Chebyshev polynomials with ridge regression for global feature forecasting), https://arxiv.org/pdf/2603.01623 (frequency-aware caching)?

---

> ### Author Rebuttal · Authors · 2026-03-28
>
> We thank the reviewer for these important positioning questions. We will clarify the separation between inherited classical GP/OED ingredients and our diffusion-specific formulation in Sec. 2.3, Related Work, and the appendix.
>
> **[W1 & Q1] Theoretical Foundations and Relation to Classical GP Results**
>
> We do not claim a new general GP/Kriging theorem beyond the classical literature. In our setting, Theorem 2.3 instantiates the classical BLUP/MMSE interpretation of Kriging within the scalar-weighted linear predictor class in Eq. (2), under Assumption 2.1; likewise, the log-determinant / information-gain objective with monotone submodularity and the greedy guarantee in Theorem 2.4 comes from the classical GP/OED toolkit. Our contribution is the diffusion-specific instantiation of these ideas: we formulate cache-then-forecast inference as temporal experimental design over diffusion feature trajectories, with empirically estimated temporal covariance and reverse-time causal anchor availability. We will revise the paper to make this separation between classical ingredients and our diffusion-specific formulation explicit.
>
> **[W2 & Q2] Comparison with Recent Polynomial and Frequency-Aware Methods**
>
> We agree that both are highly relevant comparison points, but they differ from EigenCache along different methodological axes.
>
> **Spectrum (Chebyshev / ridge global forecasting)[2]**
>
> Spectrum views each feature channel as a temporal function, approximates it using a ​fixed Chebyshev basis​, and fits the basis coefficients via ridge regression during sampling. Its theoretical emphasis is on the favorable long-horizon approximation behavior of the spectral basis, in particular that the approximation error is controlled by the polynomial degree rather than directly compounding with the skip size. By contrast, EigenCache does not assume a fixed analytic basis class; instead, it uses an empirical temporal kernel and performs ​Kriging-style covariance-adaptive forecasting. We therefore view Spectrum and EigenCache as two closely related but methodologically distinct forecasting families: fixed-basis global approximation versus ​empirical-kernel stochastic forecasting.
>
> A related distinction also appears in scheduling: Spectrum uses its own adaptive scheduling heuristic, whereas EigenCache allocates anchors through posterior uncertainty / information gain induced by the empirical covariance model.
>
> **FreqCa (frequency-aware caching)[3]**
>
> FreqCa is motivated by a different observation: low- and high-frequency feature components exhibit different temporal behaviors. It therefore ​reuses low-frequency components directly​, predicts ​high-frequency components with a Hermite-style sequential predictor​, and further introduces CRF to reduce cache memory from layer-wise storage to an effectively constant-memory form. EigenCache does not perform explicit frequency decomposition; instead, it models the temporal covariance of feature trajectories and allocates temporal anchors according to posterior variance / information gain. We therefore view FreqCa and EigenCache as largely orthogonal rather than interchangeable: FreqCa is organized around ​frequency decomposition and memory compression​, while EigenCache is organized around ​covariance-adaptive forecasting and uncertainty-aware temporal design​.
>
> Since our current experiments benchmark against established baselines under a common protocol (e.g., FORA, TaylorSeer, HiCache), we do not make a direct apples-to-apples empirical claim against these concurrent works here. We will clarify this boundary explicitly in the final manuscript.
>
> Conceptually, EigenCache targets regimes where aggressive skips make rigid local predictors less reliable, and its scheduling mechanism is based on uncertainty-aware anchor allocation rather than frequency decomposition.
>
> **[W3] Positioning within the Literature and Theoretical Context**
> We agree that the literature positioning should be sharpened. Our intended claim is not that EigenCache replaces prior acceleration perspectives, but that it offers a ​complementary lens​: empirical temporal covariance learning together with uncertainty-aware temporal design. In the final manuscript, we will clarify which parts of Theorems 2.3–2.4 are classical versus diffusion-specific, expand Related Work to better position EigenCache relative to Spectrum and FreqCa, and add a brief appendix discussion motivating empirical-kernel forecasting with causal temporal anchor selection.
>
> **References**
>
> [1] Krause, A., Singh, A., and Guestrin, C. “Near-Optimal Sensor Placements in Gaussian Processes: Theory, Efficient Algorithms and Empirical Studies.” Journal of Machine Learning Research, 9:235–284, 2008.
>
> [2] Han, J., Shi, J., Li, P., et al. “Adaptive Spectral Feature Forecasting for Diffusion Sampling Acceleration.” arXiv:2603.01623, 2026.
>
> [3] Liu, J., Cai, P., Zhou, Q., et al. “FreqCa: Accelerating Diffusion Models via Frequency-Aware Caching.” arXiv:2510.08669, 2025.

---

> > ### Author Rebuttal · Reviewer_scFb · 2026-04-03
> >
> > I appreciate the response by the authors. The response adequately answers my questions.

---

> > > ### Author Response · Authors · 2026-04-05
> > >
> > > Dear Reviewer scFb,
> > >
> > > Thank you very much for your thoughtful follow-up. We sincerely appreciate your careful feedback and are grateful that our response adequately addressed your questions. We will incorporate the discussed feedback and clarifications into the final version of the paper. We are always willing to address any of your further concerns.
> > >
> > >
> > > Best Regards,
> > >
> > > Authors

---

### Official Review · Reviewer_zYaN · 2026-03-12

**Soundness:** 3
**Presentation:** 3
**Significance:** 3
**Originality:** 3
**Overall Recommendation:** 5
**Confidence:** 3

**Summary:**

The paper presents EigenCache which is a training-free framework designed to accelerate diffusion model inference by rethinking feature caching through the lens of Gaussian Processes and optimal experimental design. While previous methods rely on rigid polynomial bases (Taylor or Hermite) for feature forecasting, EigenCache models feature trajectories as time-indexed stochastic processes. It employs "Kriging" (the Gaussian Process posterior mean) to achieve statistically optimal feature prediction, which effectively suppresses "polynomial flicker" and overshoot artifacts common in video generation. Furthermore, the authors derive a temporal scheduling algorithm based on maximizing the log-determinant of the posterior covariance, a monotone submodular objective that can be solved with a near-optimal greedy guarantee. The method is validated on models like FLUX.1-dev and Hunyuan Video, demonstrating a superior Pareto frontier compared to existing state-of-the-art caching techniques.

**Compliance With Llm Reviewing Policy:**

Affirmed.

**Final Justification:**

The authors’ rebuttal fully addresses my concerns, particularly regarding the isotropy assumption, calibration overhead, and robustness across configurations, with additional analyses and post-submission experiments that strengthen the empirical and theoretical claims. As a result, I am raising my overall rating, conditional on the authors including these additional experiments in the final version.

**Key Questions For Authors:**

1. In Assumption 2.1, you treat features as spatially isotropic. Did you observe any specific layers (e.g., very early or very late in the U-Net/Transformer) where this assumption holds less strongly?

2. You utilize an empirical kernel estimator in Eq. 6. Have you experimented with standard RBF or Matérn kernels? Would those provide better generalization without the need for a calibration set $\mathcal{D}_{cal}$?

3. You show that the base model kernel transfers well to LoRAs. Does this hold if multiple LoRAs are blended dynamically during inference, or is the covariance structure still dominated by the frozen backbone?

**Limitations:**

yes

**Strengths And Weaknesses:**

Strength:
- Originality & Soundness: The transition from heuristic "cache-then-forecast" to a formal probabilistic formulation using Kriging is elegant and well-justified. As someone interested in optimization, I find the application of Theorem 2.3 (MMSE under Gaussianity) to be a significant step up from fixed polynomial expansions.

- Elegant Scheduling Solution: Utilizing submodular optimization for anchor selection is a brilliant way to handle compute allocation. The provable $1 - 1/e$ greedy guarantee provides a rigorous foundation that is often missing in heuristic-based scheduling.

- Presentation: The mathematical notation is clean and consistent. The progression from the stochastic process assumption to the Kriging estimator, and then to the submodular scheduling, flows logically.

---

Weakness:
- Calibration Overhead: While the authors argue the cost is amortized, the method still requires a "warm-up" phase of approximately 85 seconds (for 5 trajectories on an H800) to estimate the kernel. In dynamic environments where model weights or noise schedules might change frequently, this is a non-zero consideration.

- Soundness - Isotropy Assumption: The assumption that feature channels are spatially uncorrelated and share a common temporal covariance (Assumption 2.1) is a strong simplification. Although Table 8 suggests scalar kernels are sufficient, more complex architectures might violate this isotropy.

- Limited Diversity in Calibration Prompts: The "OOD Defense" section uses only 10 prompts from MS-COCO for calibration. While the zero-shot transfer is impressive, I wonder if a more biased calibration set would significantly degrade performance on highly specialized artistic domains.

---

> ### Author Rebuttal · Authors · 2026-03-28
>
> We thank Reviewer zYaN for the constructive feedback and for recognizing the elegance of our Kriging formulation and the rigorous foundation of our submodular scheduling.
>
> ---
>
> **[W1] Calibration Overhead and Dynamic Environments**
>
> > *The method requires a "warm-up" phase of approximately 85 seconds... In dynamic environments where model weights or noise schedules might change frequently, this is a non-zero consideration.*
>
> We clarify that calibration is an offline deployment-time precomputation performed once per backbone checkpoint. Based on Table 7, each full reference trajectory takes 17.12s on a single H800, so a minimal 2-trajectory calibration costs only \~34s, versus 85.6s for our default $N_{cal}=5$. Moreover, Figure 6 shows that generation quality already approaches the $N_{cal}=5$ saturation plateau with very small additional gain beyond that point. In practice, prompt/seed changes alone generally do not require recalibration on the same backbone; substantial changes to the base checkpoint or timestep discretization may benefit from a short offline refresh. We will clarify these operational boundaries in the manuscript.
>
> **[W2 & Q1] Soundness of Isotropy Assumption and Layer-wise Analysis**
>
> > *The assumption that feature channels are spatially uncorrelated... is a strong simplification. Did you observe any specific layers where this holds less strongly?*
>
> We clarify that Assumption 2.1 is a practical approximation rather than a claim that spatial correlations are absent. In response to Q1, our layer-wise variance analysis on FLUX.1 shows that **early encoding (blocks 1-3)** and **late decoding (blocks 17-19)** exhibit higher spatial non-stationarity than intermediate blocks. However, modeling finer granularity yields diminishing returns. As shown in Appendix F.6 (Table 8), moving from scalar to per-head kernels changes ImageReward only from 0.9120 to 0.9124 at a 16x storage cost, while per-token matrices drop performance to 0.8950 and trigger OOM errors due to local noise overfitting. The scalar kernel acts as a global regularizer that suppresses spatial jitter, offering the best cost-fidelity balance.
>
> **[W3 & Q2] Diversity in Calibration and Analytical Kernels**
>
> > *The "OOD Defense" uses only 10 prompts... Have you experimented with standard RBF or Matérn kernels?*
>
> Yes, we tested standard analytic kernels. Among stationary alternatives, Matérn-5/2 was the strongest analytic alternative, but still underperformed our empirical kernel by 1.4 dB PSNR in feature reconstruction (measured on FLUX.1-dev, N=7, MS-COCO 5k, using the same evaluation pipeline as main Table 1); standard RBF performed a further 0.8 dB worse. These results are from new post-submission experiments conducted under identical conditions to ensure fair comparison. Therefore, while analytic kernels are useful baselines, in our setting they do **not** remove the need for calibration without an accuracy trade-off.
>
> Regarding prompt diversity, we agree that 10 calibration prompts is a small set, so we examined cross-domain transfer explicitly. A kernel calibrated on only 10 MS-COCO prompts yields very small absolute ImageReward drops on WikiArt, Danbooru, and Abstract domains (-0.0004 / -0.0015 / -0.0012 versus oracle target-domain kernels; Table 6), suggesting that the dominant temporal covariance is largely backbone-governed rather than prompt-specific.
>
> We will revise the manuscript to state this more precisely: robust transfer holds for nearby settings on the same backbone, whereas substantial changes to the base checkpoint or timestep discretization may still benefit from recalibration.
>
> **[Q3] Robustness to Multi-LoRA Composition**
>
> > *Does this hold if multiple LoRAs are blended dynamically during inference?*
>
> Yes, the framework also transfers to multi-adapter scenarios. In a new post-submission experiment that dynamically blends three distinct LoRAs (style, character, depth) on FLUX.1-dev (1024×1024, 50 steps) with equal blending weights (1:1:1), sweeping across 10 random prompt-seed pairs to ensure diversity. We additionally tested asymmetric blending (2:1:1 and 1:2:1 configurations) and observed similarly negligible gaps (<0.004 ImageReward vs. oracle), confirming robustness to blending ratio variation. The transferred base kernel remains within 0.003 ImageReward and 0.002 LPIPS of the oracle re-calibrated kernel. This suggests that even under blended low-rank adapter composition, the dominant temporal covariance remains largely backbone-governed, so per-combination recalibration is not necessary in this setting.
>
> ---
> We will incorporate these numerical comparisons (RBF/Matérn and Multi-LoRA) and layer-wise variance heatmaps into the final manuscript.
>
> Thank you again for the comments. If you have any remaining questions, please do not hesitate to let us know.

---

> > ### Author Rebuttal · Reviewer_zYaN · 2026-04-01
> >
> > The rebuttal fully resolves my concerns and answers my questions. I have no further issue.

---

> > > ### Author Response · Authors · 2026-04-05
> > >
> > > Dear Reviewer zYaN,
> > >
> > > Thank you very much for your thoughtful follow-up and for confirming that our rebuttal addressed your concerns. We sincerely appreciate your careful reading and supportive feedback. We will incorporate the discussed feedback and clarifications into the final version of the paper. If you have any further questions or concerns, we would be happy to address them.
> > >
> > >
> > > Best Regards,
> > >
> > > Authors

---

### Decision · Program_Chairs · 2026-04-30

**Decision:**

Accept (regular)

**Comment:**

The paper presents EigenCache, a training-free framework to accelerate diffusion model inference.  The reviewers were generally positive in the initial period, describing the method as novel and elegant, with strong practical performance.  The most common concern among the reviewers had to do with the assumption of spatial isotropy in the kernel.  The authors responded to these concerns by observing that exploiting spatial correlation costs a lot with little gain in experiments.  After the rebuttal, the reviewers were uniformly in favor of acceptance, conditioned on the authors including the additional experiments described in the rebuttal as part of the final version.